

SciPost Phys. Comm. Rep. 11 (2025)

# Visions in quantum gravity

Luca Buoninfante[1*], Benjamin Knorr[2†], K. Sravan Kumar[3‡], Alessia Platania[4°],
Damiano Anselmi[5], Ivano Basile[6], N. Emil J. Bjerrum-Bohr[4], Robert Brandenberger[7],
Mariana Carrillo González[8], Anne-Christine Davis[9], Bianca Dittrich[10],
Paolo Di Vecchia[11], John F. Donoghue[12], Fay Dowker[8,10], Gia Dvali[13], Astrid Eichhorn[2],
Steven B. Giddings[14], Alessandra Gnecchi[15], Giulia Gubitosi[16,17], Lavinia Heisenberg[2],
Renata Kallosh[18], Alexey S. Koshelev[19], Stefano Liberati[20,21,22], Renate Loll[1],
Leonardo Modesto[23], Paulo Moniz[24], Daniele Oriti[25,26], Olga Papadoulaki[27],
Jan M. Pawlowski[2], Roberto Percacci[20], Lesław Rachwał[28], Mairi Sakellariadou[29],
Alberto Salvio[30,31], Kellogg Stelle[8], Sumati Surya[32], Arkady Tseytlin[8],
Neil Turok[33], Thomas Van Riet[34] and Richard P. Woodard[35]

★ luca.buoninfante@ru.nl , † knorr@thphys.uni-heidelberg.de ,
‡ sravan.kumar@port.ac.uk , ○ alessia.platania@nbi.ku.dk

## Abstract

To deepen our understanding of Quantum Gravity and its connections with black holes and cosmology, building a common language and exchanging ideas across different approaches is crucial. The Nordita Program *"Quantum Gravity: From gravitational effective field theories to ultraviolet complete approaches"* created a platform for extensive discussions, aimed at pinpointing both common grounds and sources of disagreements, with the hope of generating ideas and driving progress in the field. This contribution summarizes the twelve topical discussions held during the program and collects individual thoughts of speakers and panelists on the future of the field in light of these discussions.

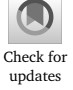

---

1 High Energy Physics Department, Institute for Mathematics, Astrophysics,
and Particle Physics, Radboud University, Nijmegen, The Netherlands
2 Institute for Theoretical Physics, Heidelberg University,
Philosophenweg 12&16, 69120 Heidelberg, Germany
3 Institute of Cosmology and Gravitation, U. Portsmouth, Dennis Sciama Building,
Burnaby Road, UK, Portsmouth, PO1 3FX, UK
4 Niels Bohr International Academy, The Niels Bohr Institute,
Blegdamsvej 17, DK-2100 Copenhagen Ø, Denmark
5 Dipartimento di Fisica "Enrico Fermi", Università di Pisa,
Largo B. Pontecorvo 3, 56127 Pisa, Italy, and INFN,
Sezione di Pisa, Largo B. Pontecorvo 3, 56127 Pisa, Italy
6 Arnold-Sommerfeld Center for Theoretical Physics,
Ludwig Maximilians Universität München,
Theresienstraße 37, 80333 München, Germany
7 Department of Physics, McGill University, Montréal, QC, H3A 2T8, Canada
8 Theoretical Physics, Blackett Laboratory, Imperial College, London, SW7 2AZ, UK

**9** Department of Applied Mathematics and Theoretical Physics, University of Cambridge,
Wilberforce Road, Cambridge, CB3 0WA, UK

**10** Perimeter Institute, 31 Caroline Street North, Waterloo ON, N2L 2Y5, Canada

**11** Nordita, KTH Royal Institute of Technology and Stockholm University,
Hannes Alfvéns väg 12, SE-11419 Stockholm, Sweden

**12** Department of Physics, University of Massachusetts, Amherst, MA 01003, USA

**13** Max-Planck-Institut für Physik, Föhringer Ring 6, 80805 Munich, Germany

**14** Department of Physics, University of California, Santa Barbara, CA 93106, USA

**15** INFN, Sezione di Padova, Via Marzolo, 8, 35131 Padova, Italy

**16** Dipartimento di Fisica Ettore Pancini, Università di Napoli Federico II

**17** INFN, Sezione di Napoli, Complesso Univ. Monte S. Angelo, I-80126 Napoli, Italy

**18** Stanford Institute for Theoretical Physics and Department of Physics,
Stanford University, Stanford, CA 94305, USA

**19** School of Physical Science and Technology, ShanghaiTech University,
201210 Shanghai, China

**20** Scuola Internazionale Superiore Studi Avanzati (SISSA),
Physics Area, Via Bonomea 265, 34136 Trieste, Italy

**21** Institute for Fundamental Physics of the Universe (IFPU), Via Beirut 2, 34014 Trieste, Italy

**22** INFN, Sezione di Trieste, Via Valerio 2, 34127 Trieste, Italy

**23** Dipartimento di Fisica, Università di Cagliari,
Cittadella Universitaria, 09042 Monserrato, Italy

**24** Departamento de Física, Centro de Matemática e Aplicações (CMA-UBI),
Universidade da Beira Interior, Rua Marquês d'Avila e Bolama,
6200-001 Covilhã, Portugal

**25** Depto. de Física Teórica, Facultad de Ciencias Físicas,
Universidad Complutense de Madrid,
Plaza de las Ciencias 1, 28040 Madrid, Spain

**26** Munich Center for Quantum Science and Technology (MCQST),
Schellingstrasse 4, 80799 München, Germany

**27** CPHT, CNRS, École polytechnique, Institut Polytechnique de Paris,
91120 Palaiseau, France

**28** Departamento de Física, ICE, Universidade Federal de Juiz de Fora,
Campus Universitário - Juiz de Fora, 36036-330, Minas Gerais, Brazil

**29** Theoretical Particle Physics and Cosmology Group, Physics Department,
King's College London, University of London, Strand, London WC2R 2LS, UK

**30** Physics Department, University of Rome Tor Vergata,
via della Ricerca Scientifica, I-00133 Rome, Italy

**31** INFN - Rome Tor Vergata, via della Ricerca Scientifica, I-00133 Rome, Italy

**32** Raman Research Institute, CV Raman Ave, Sadashivanagar, Bangalore, 560080, India

**33** Higgs Centre for Theoretical Physics, James Clerk Maxwell Building,
Edinburgh EH9 3FD, UK

**34** Instituut voor Theoretische Fysica, K.U. Leuven,
Celestijnenlaan 200D, B-3001 Leuven, Belgium

**35** Department of Physics, University of Florida, Gainesville, FL 32611, USA

## Contents

# 1 Quantum gravity: From gravitational effective field theories to ultraviolet complete approaches

A vast amount of observational data has confirmed the theoretical predictions of General Relativity (GR), making it the currently best known theory to describe classical aspects of the gravitational interaction, from cosmological to sub-millimeter scales. Despite this phenomenal success, however, there are still fundamental questions that remain unanswered. At small scales, GR predicts the existence of singularities in black holes (BHs) and at the big bang, where spacetime ends and the theory breaks down. At the quantum level, Einstein's theory lacks predictivity in the ultraviolet (UV) regime, i.e., at high energies, as it is plagued by UV divergences that cannot be absorbed in a finite number of free parameters. It is generally believed that a consistent theory of quantum gravity (QG), valid at all energy and distance scales, is needed to deal with these challenges.

The question of how to formulate such a consistent theory has grabbed great interest and given rise to longstanding debates. Indeed, in the past decades, several promising attempts and novel ideas have been proposed [1, 2]. These include perturbative and non-perturbative approaches based on the framework of quantum field theory (QFT) such as local [3–5] and non-local [6] higher-derivative theories of gravity, asymptotically safe quantum gravity (ASQG) [7,8] and causal dynamical triangulations (CDT) [9], but also other proposals based on alternative frameworks such as string theory (ST) [10,11], loop quantum gravity (LQG) [12], spin foams [13], group field theory (GFT) [14], causal set theory (CST) [15], and non-commutative geometry [16].

**Overview and objectives of the program.** The overall scope of the Nordita Scientific Program *"Quantum Gravity: From gravitational effective field theories to ultraviolet complete approaches"* was to achieve a thorough assessment of the current status of QG by providing a critical inspection of the basic issues, and discussing several attempts and approaches to the quantization of gravity, including the most recent proposals. The program ran for four weeks: The first week consisted of the PhD school *"Towards Quantum Gravity"* [17], while the other three weeks featured workshops where theoretical and phenomenological aspects of several approaches to QG were intensely discussed. Different schools of thoughts were brought together, in such a way that everyone could benefit from fruitful discussions and learn from each other. In addition to individual talks, twelve panel discussions were organized with the aim of serving as a platform of constructive debates and new insights. To stimulate the discussions and make the meeting more productive, specific topics and concrete questions were prepared for each of the sessions. Panel chairs and panelists were appropriately chosen in order to trigger discussions on questions where different approaches are in conflict, and to potentially arrive at a better understanding of the implicit assumptions behind every statement.

**Importance of the program.** Over the past decades, the effective field theory (EFT) community as well as communities exploring various approaches to QG have worked independently, focused on different issues, and thereby achieved milestones in different aspects. This scientific program brought all these communities together, allowing for ample time to share viewpoints and achievements, raise problems, and exchange potential solutions. Old and new questions were addressed by exploiting the background knowledge and achievements of different communities, which facilitated a cross-fertilization among various approaches. With the recent experimental successes of high precision observations of the cosmic microwave background (CMB) [18], the direct detection of gravitational waves (GWs) [19] and the construction of BH shadows [20], we are closer than ever before to probe deviations from GR which potentially originate from quantum effects. Therefore, it is of utmost importance to determine what quantum-gravitational effects can look like according to different theories, and in which observables in the infrared (IR) they are most likely to be detected. In this respect,

the program and its scope were very timely.

It is important to emphasize that the innovative aspect of this program was its scientific diversity. Every QG proposal has its own point of view on physics at fundamental scales. Most of the existing approaches were discussed during this program. This Nordita Scientific Program, with its holistic and agnostic view, brought new momentum and perspectives to future research in QG, ranging from theoretical formal aspects to phenomenology.

All lectures, talks, open discussions and panel discussion of the program were recorded, and are available on YouTube on the channel @Quantumgravity.nordita. This contribution summarizes the main highlights of the panel discussions and collects individual thoughts by invited speakers and panelists in the light of the talks and discussions. It is organized as follows.

**Sec. 2:** The main highlights and outcomes of each of the twelve topical panel discussions are summarized by the organizers.

**Sec. 3:** The individual thoughts of invited speakers and panelists who participated in the second and third week of the program are collected. During these two weeks, the topical discussions focused on *"Formal aspects and consistency of Quantum Gravity approaches"*.

**Sec. 4:** The individual thoughts of invited speakers and panelists who participated in the fourth week of the program are collected. During this week, the topical discussions focused on *"Quantum Gravity phenomenology"*.

**Sec. 5:** Conclusions and final thoughts by the organizers are drawn.

With this contribution, the organizers would like to provide a snapshot of the state-of-the-art, open questions, and visions of different QG approaches, highlighting sources of disagreement, clarifying misunderstandings, and pinpointing common grounds. We hope that this work will serve as a stepping stone to generate new ideas, encourage more frequent discussions between different communities, and ultimately reach consensus about some of the subtleties and generic features of QG.

*The organizers Luca Buoninfante, Benjamin Knorr, K. Sravan Kumar, and Alessia Platania became aware of the fact that one of the speakers and panelists, **Jerzy Lewandowski,** sadly passed away shortly after the program. Despite his failing health, he went out of his way and actively contributed to our program. We are honored to have had him in our program, and we would like to take this opportunity to acknowledge his great contributions to science.*

## 2 Summarizing discussions: Common questions from different viewpoints

### 2.1 Quantum field theory framework for quantum gravity: Yes or no?

> **Chair:** Ivano Basile
>
> **Panelists:** Damiano Anselmi, Bianca Dittrich, Paolo Di Vecchia, Roberto Percacci, Thomas Van Riet
>
> **Recording:** https://youtu.be/BCxoljWEc64

This panel discussed the question of whether the framework of QFT is sufficient to describe gravity at all scales, or whether we need to go beyond it. Panelists had different perspectives on this question, from advocacy of QFT as a predictive and simple framework, to more cautious stances emphasizing its limitations. Discussions touched on the following key points:

**EFT as a common ground and its limitations.** One emerging viewpoint, shared by researchers working on ST and QFT-based approaches, was the statement that EFT should be the commonly accepted framework and an "anchor" for research in QG. Percacci voiced that gravitational EFT should already be considered a theory of QG (cf. section 3.18) — though in EFT gravity is a force (mediated by the graviton) and this feature seems not to match the viewpoint of most discrete approaches (this was further discussed in several open and panel discussions, see e.g. Panel 6). Di Vecchia pointed out that the framework is also successfully used to extract classical observables, particularly in GW physics, where the production of GWs from a BH merger at large impact parameter can be computed using EFT/QFT tools, like scattering amplitudes [21]. Nonetheless, the framework also has its limitations, and different approaches can greatly differ when going beyond this common ground.

**QFT for QG?** The core question of the panel — whether the QFT framework is sufficient to describe QG — was a topic of contrasting opinions. In going beyond EFT, QFT approaches are certainly the most conservative. Percacci emphasized that, to a certain extent, also CDT, GFT, and LQG can be considered as QFTs — just discretized ones — but that the continuum limit is essential for predictivity. He also pointed out that field theory should not be deemed "dead", recalling times it was previously dismissed only for new insights to reinvigorate it (e.g., quantum electrodynamics (QED), the Higgs mechanism, strong interactions). Anselmi also strongly supported QFT, calling it the "highest achievement of the human mind" and asserting that a local QFT is still compatible with data, and suffices for QG, with appropriate relational observables. Other panelists though voiced that defining QG observables might require us to go beyond QFT (see below). In addition, Basile stated that UV/IR mixing should be a generic feature of QG, which likely cannot be described within the framework of QFT, unless, for instance, an infinite number of fields is considered. That would however go in the direction of string field theory, as pointed out by Percacci. In this context, Van Riet argued that ST offers valuable lessons about EFT in the IR: Understanding basic models, even those that are not directly physically realized (in analogy with the Ising model), is essential for tackling more complex physical systems. In this sense, independent of whether ST is realized in nature, it could be used as a "learning tool" to constrain EFT.

**QG observables and defining QFTs.** Observables were a recurring topic in highlighting the differences between QFT-based and beyond-QFT approaches. The nature of observables in QG was thus touched on multiple times during the discussion and diverse viewpoints emerged. Various panelists supported the idea that scattering amplitudes are the only observables that can be reasonably defined in QG. Whether there are other observables beyond scattering amplitudes emerged as an open question, but there was an emerging agreement on the fact

that observables should be relational [22]. Related to this, it was pointed out by Dittrich that the question of the panel "QFT framework for QG: Yes or no?" may be ill-posed, because one needs to specify what QFT is in the first place: QFT on a fixed background provides local observables only, but a general QG theory would require relational observables. An ongoing exchange saw Dittrich and Hassan propose that QG might first need to be specified before discussing observables. In particular, if relational observables are local, a physical reference system is essential. Nonetheless, it was also remarked that probing observables beyond the Planck scale might be unachievable due to classicalization effects [23].

**Beyond EFT — Semiclassical methods as a bridge to QG and the path integral debate.** Semiclassical methods were discussed as a potential bridge between the EFT and QG regimes. Van Riet specifically remarked that the form of the Bekenstein-Hawking entropy (including the exact numerical coefficient) places strong indirect constraints on QG [24] — a viewpoint also shared by other participants. A theory of QG should be able to reproduce not only the area scaling, but also the coefficient in front of it. The saddle point method, while powerful, does not know about the microscopic details of the theory (aside from the implicit sum over topologies involved in the sum over the saddles). Some participants opinioned that it remains unsettled which saddles are crucial. Di Vecchia added that Wick rotations in saddle point approximations are acceptable, but cautioned that Hartle-Hawking boundary conditions might lose relevance if QG is fundamentally Lorentzian. Path integrals and the sum over topologies were also areas of debate, with panelists expressing divergent views on the saddles and on the role of the spacetime signature in identifying them. This uncertainty is linked to Turok's point in Panel 4, that identifying the right saddle points is guesswork.

In conclusion, the panel saw an agreement about the crucial role of EFT as a common ground for research in QG. Going beyond it, however, two diverging lines of thought emerged. One questioned if QFT is adequate to address core challenges in QG, such as BH microstate counting and relational observables, proposing that they reflect a deeper theoretical crisis requiring us to go beyond QFT. The alternative viewpoint was that QFT in the past has overcome several challenges, and QG is just the next one. The discussion highlighted key open questions, from observables to semiclassical approximations, topology changes, and BH entropy. The future of QG research appears to rest on either adapting QFT to meet these challenges or conclusively going beyond it.

## 2.2  Unitarity, causality, stability

> **Chair:** Roberto Percacci
>
> **Panelists:** Ivano Basile, Bianca Dittrich, John F. Donoghue, Steven B. Giddings, Alexey Koshelev
>
> **Recording:** https://youtu.be/dY0XCN6HFBk

Current experiments are not in contradiction with certain physical requirements that one often demands in order to constrain theories in low-energy regimes, for example below the inflationary scale of $10^{14}$ GeV. Key physical requirements are those of (i) unitarity, (ii) causality and (iii) stability.

(i) Unitarity means that quantum probabilities are conserved, and it is usually formulated as a condition on the evolution operator [25]. Its formulation in terms of the S-matrix, in most cases, requires the presence of an asymptotically flat spacetime.

(ii) Causality can be phrased by saying that no faster-than-light communication can occur, within the limits set by the Heisenberg uncertainty principle. The concept of causality can be formulated in various inequivalent ways [26]: For example, via off-shell conditions such as local commutativity (also known as microcausality) and the more general Bogoliubov causality condition [25], or via on-shell requirements such as the absence of resolvable Shapiro time advances [27]. Causality often translates into analyticity conditions on the S-matrix.

(iii) Stability means that no *unphysical* instabilities should be present [28]: For example, the vacuum state of the Hamiltonian must exist and be stable, tachyonic and ghost-like degrees of freedom should be absent or harmless [29], the classical evolution should not be plagued by runaway solutions on attainable time scales. Basile nicely summarized the concept of physical stability with the statement *"stuff can exist"*.

While these requirements may be valid and sufficient to constrain low-energy physics, they could be challenged by a QG approach that aims to obtain a complete description of gravity and its coupling to matter at arbitrary high-energy and short-distance scales. In this panel discussion, different viewpoints were shared by the panelists and other participants.

**Unitarity and BH evaporation.** It was remarked that the lack of a full understanding of the endpoint of BH evaporation makes it unclear what happens to the information that is thrown into a BH. Giddings pointed out that the notion of locality apparently must be given up to ensure that information is not lost and that evolution is kept unitary [30–32]. Since QG effects should become important towards the end of BH evaporation, the above reasoning might imply that some form of non-locality is a generic feature of any QG approach.

**Does unitarity make sense in QG?** On the one hand, from an S-matrix perspective, unitarity and causality are clearly defined by the properties of scattering amplitudes. On the other hand, Dittrich raised the point that it is not clear whether a unitary evolution could make sense in QG [33]. She suggested that the notion of unitarity could be replaced by that of isometry, which only demands that the inner product between any two states is preserved, while the actual number of states can increase [34]. This idea could also be supported by the fact that the expansion of the universe could imply that the size of the Hilbert space increases with time and, hence, that the unitarity requirement could be too strong. Dittrich also noted that in the canonical approach to QG, one has to deal with the Wheeler-DeWitt equation where no explicit time coordinate appears. This might suggest that the notion of time evolution has to be constructed via relational observables by first identifying a set of clocks and rods, and then defining a *physical* Hamiltonian [35, 36].

**Unitarity and ghosts.** Concerns about the unitarity of the S-matrix were raised in the context of the QFT framework when considering a perturbatively renormalizable completion of GR that consists in the inclusion of quadratic curvature operators in the Lagrangian (quadratic gravity) [3–5, 37, 38]. In this case, the renormalizability property is recovered at the cost of introducing a massive spin-two ghost field. Especially in the last decade, several ideas have been proposed to handle these ghosts [3–5, 39]. What emerged from the discussion is that unitarity and perturbative renormalizability in QG can be reconciled if and only if something else is given up such as causality or locality, as especially pointed out by Donoghue and Koshelev. Koshelev particularly emphasized that the requirement of unitarity (in the sense of the absence of ghosts) is much more important than that of causality and locality, which might be lost in QG.

**Topology change in QG?** Basile noted that in theories where topology change is allowed, gravitational anomalies involving it (known as Dai-Freed anomalies [40, 41]) could occur, and eventually lead to unitarity violation. These go beyond the well-known (large) diffeomorphism anomalies [42]. Therefore, the requirement of unitarity demands the cancellation of this type

of anomalies. Luckily, these are known to cancel in the Standard Model of Particle Physics (SM) coupled to gravity. More general questions were raised on whether topology should fluctuate in QG, e.g., whether in a path-integral, one should sum over all possible different topologies. There was no agreement on this point. For example, Dittrich mentioned that topology change might be important for calculating the gravitational entropy from a Lorentzian path integral; it was also pointed out that topology change is allowed in CST, but probably not in CDT. In particular, Loll stated that in the non-perturbative gravitational path integral, no one knows how to make sense of summing over topologies because of the super-exponential growth of configurations, which not even in two dimensions can be controlled uniquely. She also stressed that there is no physical imperative to include such a sum, and CDT shows that one obtains non-trivial results without it. On the other hand, Van Riet supported that topology change is important in QG, highlighting that for instance baby universes are crucial to understanding unitarity.

**Causality and non-locality.** Some agreement was reached that some kind of causality violation should occur in QG. For example, it was observed that the absence of gauge-invariant *local* observables in QG requires the need to construct *non-local* gauge-invariant operators for which the microcausality condition is then violated [43, 44]. This feature should be common to all QG approaches in which diffeomorphism invariance is respected, and should become more evident at the non-perturbative level, when spacetime fluctuations are not negligible. It was noted that some form of causality is also violated in non-local approaches to QG in which the quantum-gravitational Lagrangian contains non-polynomial form factors. These are differential operators whose specific form is appropriately chosen to avoid the presence of ghost-like degrees of freedom [45, 46]. One of the challenges in these theories is to quantify more precisely the degree of such causality violation [47, 48].

**Causality and arrow(s) of time.** In the context of quadratic gravity, Donoghue remarked that the presence of the spin-two ghost could introduce an opposite arrow of causality at energy scales larger than its mass, so that in a QFT context the very notion of causality loses its meaning at the microscopic level [5, 49]. This might suggest that causality is an emergent phenomenon, and acquires physical meaning only at macroscopic scales [50, 51], i.e., at energy scales smaller than the mass of the spin-two ghost particle. Donoghue also remarked that quadratic gravity could explain inflation thanks to the presence of an additional spin-zero degree of freedom (via Starobinsky inflation [52–54]). He also made the observation that acausal effects induced by the additional spin-two ghost could be relevant for early-universe cosmology, and eventually leave an imprint on the CMB.

**Stability.** The stability issue was discussed especially in the context of QFT-based approaches to QG, in particular in quadratic gravity [3, 5] and non-local theories of gravity [45, 46]. In both cases, while the stability of the theory seems to be under control at the perturbative level, it was remarked that it is still unclear whether instabilities can arise non-perturbatively and in the classical limit.

In summary, the panel discussion made it clear that similar problems can emerge in completely different contexts. For example, troubles with having a consistent unitary evolution appear for evaporating BHs and expanding universes, but also when demanding perturbative renormalizability for a QFT of the gravitational interaction. Furthermore, despite various disagreements on how to deal with the quantization of the gravitational interaction and whether the very concept of unitarity should make sense in QG, some agreement was reached on the idea that some form of causality and locality may be lost at the microscopic level, due to intrinsic features of QG. There was also consensus on the fact that in QG, the preservation of some form of unitarity is more relevant than the questions of (a)causality and (non-)locality.

## 2.3 Features and principles of quantum gravity: Degrees of freedom, non-locality, non-commutativity

> **Chair:** Thomas Van Riet
>
> **Panelists:** Fay Dowker, Steven B. Giddings, Alexey Koshelev, Renate Loll, Daniele Oriti
>
> **Recording:** https://youtu.be/DZSZc7qE_qE

Several QG theories have been proposed over the past decades, but it is very difficult to make comparisons between them. In this regard, it could be very useful to identify key features that can be used to discriminate between different approaches, and to understand whether there are model-independent properties that should characterize the quantum-gravitational interaction. This panel discussion debated some candidates for such universal features: Fundamental degrees of freedom, spacetime emergence, non-locality, UV/IR mixing, and non-commutativity.

**Spacetime metric: Fundamental or emergent?** One clear message which came out from the discussion is that there is no agreement on whether spacetime is fundamental, and on the fact that there is no unique way to make spacetime emergent. Some QG approaches keep the spacetime metric as fundamental (e.g., some QFT-based approaches such as ASQG [7,8], quadratic gravity [3–5, 37], non-local gravity [45, 46], CDT [9]), while others assume that the metric can be derived starting from more fundamental degrees of freedom. This second class can be divided into two groups: One that is still based on a continuum spacetime description (e.g., ST [10, 11] and the anti-de Sitter/conformal field theory (AdS/CFT) correspondence [55]), and another group which assumes that the spacetime description is fundamentally discrete (e.g., LQG [12], spin foams [13], CST [15], GFT [14]), see also Panel 6. In this context, Oriti pointed out that understanding QG and whether spacetime is emergent might require a reassessment of the foundations of quantum mechanics (QM) (specifically, the measurement problem and the role of observers, see also Panel 1). He also opinioned that in approaches in which spacetime and geometry (thus the metric) are not fundamental, we should expect that *all* properties like unitarity, locality, causality, commutativity (of spacetime observables), and the equivalence principle are emergent and approximate only, with the real question being which one emerges first or breaks down later and which one is more robust and universal as an emergent property.

**Low-energy regime, continuum limit, and gravitons.** An important question that appeared to be still unanswered in discrete approaches, but also in theories where spacetime is fundamental but the mathematical methods rely on discretization (e.g., CDT), is how to recover a continuum spacetime description and establish a connection with the low-energy regime. For example, it is unclear how to recover the notion of a graviton and the propagation of GWs in the appropriate classical limit. No satisfactory agreement was reached. On the one side, Di Vecchia, Giddings and the other participants working on QFT and stringy approaches voiced that recovering the notion of a graviton in the appropriate low-energy perturbative regime is necessary for the consistency of the theory. On the other side, Loll, Surya, and other participants working on discrete approaches critically questioned whether gravitons should be a generic feature of all QG approaches (see also section 3.20); Dowker added that at least in CST it should be possible to make contact with the standard low-energy EFT description but it is not yet clear how to take the continuum limit. In this context, Van Riet made the important observation that researchers in QG not only disagree on the underlying theory, but also on the questions that such a theory is supposed to answer: *is there an observable that all QG approaches agree should be computed, also with the scope of comparing approaches?* Many participants suggested that such observables should be scattering amplitudes. Yet, due to op-

posing ideas on the role of EFTs and the very notion of a graviton in the IR, this view was not shared by everyone. Platania however commented that EFT is simply a way to parametrize QG corrections, and that the specific combinations of Wilson coefficients entering scattering amplitudes are scheme-independent. Hence, one can even forget about scattering processes, and still use them as well-defined observables that could be compared across approaches.

**Non-locality and UV/IR mixing.** One feature that seems to be common to most of the QG approaches is that the locality of the gravitational interaction must be given up in some way or another. Some model-independent arguments in favor of specific non-localities are based on the absence of gauge-invariant local observables [43, 44], and on the mixing of UV and IR modes due to BH formation [56–58]. In particular, Giddings remarked that in a scattering process, a BH can be formed when sufficiently high energies are confined in a sufficiently small region. This reasoning implies that higher energies (UV) will only be able to probe larger distances (IR) because the BH horizon will increase in size if more energy is thrown in. Heuristic entropic arguments were used to claim that this type of BH formation can make the scattering amplitude exponentially suppressed, so that the property of polynomial boundedness (i.e., a version of locality) is violated. It was suggested that this feature may indicate that QG should be described in the framework of non-localizable QFTs [59, 60].

**Black holes and asymptotic graviton states in the UV.** Although the physical arguments supporting BH formation via graviton scattering appear to be well-motivated, rigorous computations supporting this reasoning are still lacking, and some criticisms were put forward during the discussion. In particular, Pawlowski pointed out that the first main issue to address before looking at the high-energy scattering is to define what an asymptotic graviton state is when self-interactions are not negligible [61], i.e., especially in the UV regime. Quantum chromodynamics (QCD) was used as an example since gluons are not appropriate particle states in the IR due to non-linear symmetries and strong coupling. At the end of the discussion there was partial agreement that further investigation is needed to understand whether gravitons are the right asymptotic states *in the UV*. This topic also reappeared in Panel 8.

**Non-commutativity.** The non-locality of gauge-invariant observables introduces some type of non-commutativity due to the fact that commutators of observables are non-zero for space-like separations. Basile observed that a remnant of this property can also be seen in the perturbative EFT regime [62]. Since the EFT framework is expected to be a common ground where most (if not all) QG approaches should co-exist in some appropriate low-energy regime, non-commutativity of gauge-invariant observables [43, 44] might be a generic feature of QG. This was particularly emphasized by Giddings.

**Additional discussions on a broader level.** Part of the panel also focused on what kind of additional or alternative degrees of freedom could appear at the fundamental level. For example, Dowker pointed out that in CST the microscopic entities are discrete order relations. Other participants remarked that, in addition to the metric, extra geometric degrees of freedom could appear such as connections and torsion, which could become dynamical at high energies. Furthermore, Loll emphasized the need to develop more suitable and powerful computational methods to gain a better understanding of non-perturbative aspects of QG. In this regard, she noted that, although matter and its coupling with gravity do matter, understanding the non-perturbative sector of pure gravity could already be a good starting point. Lastly, Giddings pointed out that a suitable starting point for QG may be finding the appropriate structure on the Hilbert space.

In summary, no consensus emerged from the panel discussion on whether the spacetime metric should be fundamental or not. However, there was agreement that in QFT-based approaches to QG, the spacetime metric is still fundamental and could be accompanied by additional degrees of freedom such as connections and torsion. By contrast, in different QG approaches that are not based on the QFT framework, there is room for alternative fundamen-

tal entities. Furthermore, despite the existence of heuristic arguments, most of the participants agreed that deeper investigations are needed to understand the dynamics of gravitons at high energies and the role of BH formation in a scattering process. Similarly to Panel 2, there was consensus that some form of non-locality might be an intrinsic feature of QG.

## 2.4 Status of the string paradigm

> **Chair:** Steven B. Giddings
>
> **Panelists:** Damiano Anselmi, Ivano Basile, Paolo Di Vecchia, Neil Turok, Thomas Van Riet
>
> **Recording:** https://youtu.be/lQK9CliSICI

This panel explored the relevance and limitations of ST in addressing key questions in QG, with panelists sharing a range of perspectives, from critiques of ST's dream of being a zero-parameter theory to defenses of its foundational insights. The discussion covered several central topics:

**Core challenges in QG and ST.** Giddings started the discussion by outlining two major facets of the QG problem: (i) non-renormalizability, and, more recently, (ii) issues surrounding non-unitarity, particularly in the physics of quantum BHs. He argued that further clarity is required on the nature of QG observables (a topic that turned central in several panels), the role of the Hilbert space in determining states, unification, and quantum cosmology (QC). While Giddings acknowledged ST's partial successes — such as addressing the non-renormalizability and the question of observables via the S-matrix formalism — he noted that it has not fully addressed several fundamental issues, particularly unitarity and the role of holography beyond AdS/CFT.

**Historical skepticism and the problem of predictivity.** Anselmi expressed his longstanding skepticism of ST, tracing its ideological evolution from the 1950s' focus on S-matrix principles, to the 1995 superstring revolution. In his view, ST's ambition to provide a "theory without parameters" was an overreach, and he highlighted that some proponents dismissed criticism by claiming that ST could ultimately predict everything. He criticized what he saw as a trend within ST towards "quantum denial", referring to the treatment of ST largely as a classical theory (without string loop corrections). He noted that some in the ST community prioritized aesthetic qualities over empirical grounding. This criticism was acknowledged, but it has also been noted that the string community has gone beyond that state: Claims are formulated more carefully, and quantum corrections are being taken into account. A consensus emerged in the discussion, that today's researchers in ST should not be held accountable for overconfident claims made decades ago by ST's pioneers.

**ST and its misconceptions.** Basile supported Giddings' view that EFT serves as a common foundation in QG — though this fact is not shared across all approaches (see also Panel 1, Panel 3 and Panel 11). ST seems to provide a consistent completion of all interactions, it points to a universal UV/IR mixing due to gravity, and has a very rigid structure: It unavoidably includes gravity and Yang-Mills theories, holography is built-in into the theory, and its UV behavior is universal. Basile argued that ST has the potential to provide universal insights at high energies, even if the IR predictions — aka, phenomenology — remain challenging. He addressed several common misconceptions: ST does not necessarily predict supersymmetry or extra dimensions (it only does so in the simplest sectors), it has a countable (likely finite) number of EFTs at a fixed cutoff, it has specific predictions, for instance, the high-energy scattering of gravitons [63], and it includes non-perturbative insights [64], contrary to critics suggesting otherwise. He concluded that ST might not support eternal de Sitter (dS) spaces,

only metastable ones, suggesting dynamical dark energy as a genuine prediction of the theory.

**ST as an extension of QFT.** Di Vecchia argued that ST is effectively an extension of QFT, in which string interactions yield gauge and gravitational interactions as a low-energy limit. He highlighted the uniqueness of ST in 10 dimensions, noting however the challenge of obtaining four-dimensional models through compactifications. He defended ST's lack of predictivity in collider physics, comparing it to the SM, which also does not predict (all) fundamental constants from first principles. Di Vecchia highlighted that high-loop beta functions are easier to compute in ST than in QFT, emphasizing ST's role as a computational tool.

**Experimental realism and a shift in focus.** Turok shared his evolving view of ST, recounting an early optimism that faded over time. He criticized ST's detachment from observational reality, especially regarding cosmology, and argued for a research focus on simpler, empirically grounded theories. He credited ST with advancing theoretical tools, but asserted that practical progress in QG would likely come from more minimalistic frameworks, which, while less ambitious, might better capture essential features of our universe.

**Landscapes and predictivity.** Van Riet addressed criticisms of ST's "landscape" of possible vacua, arguing that this vast landscape does not inherently lack predictivity: The parameter space of QFTs, and in particular of the SM is much bigger. One key example is QCD: The fundamental Lagrangian is simple, but the low-energy vacuum structure is very complicated. The swampland program [65–69] suggests instead that almost nothing goes: When compactifying, one sees that it is almost impossible to get out arbitrary models. Because of how EFTs and the decoupling mechanism work, it is natural that QG effects are Planck-mass suppressed — though potentially visible through indirect observables. For this reason, building a QG theory (not specifically ST) requires exploiting internal logical consistency, which imposes much stronger constraints than observational consistency. An example is the area law, which QG should reproduce from first principles — a task where ST claims success [70, 71].

**Additional discussions on a broader level.** The discussion highlighted different perspectives on the status of ST. A key question is whether ST is unitary: While ST is formulated to be perturbatively unitary, the question of non-perturbative unitarity, particularly in the context of BHs, is unsettled. This distinction led to a broader examination of the nature of unitarity in ST specifically, and QG in general (see also Panel 2). It was also emphasized that the path to the widespread acceptance of a theory can often be protracted, as it has happened for Yang-Mills theories. In this context, Anselmi highlighted the importance of empirical validation, urging the community to demonstrate their theories through concrete calculations and comparison with observations. In addition, Oriti reflected on the strong claims that were historically made about ST's predictive power. He cautioned against the idea that ST could be a one-size-fits-all solution, suggesting that there may be numerous microscopic models that, once coarse-grained, reveal universal features applicable across different approaches to QG. Then one should advocate for the identification of common features among the various frameworks rather than establishing the uniqueness of ST. This, however, was pointed out to be the core idea of the swampland program: Identifying what goes in QG. This led the focus to a particular swampland conjecture, the no dS conjecture: Various hints indicate that eternal dS space may not exist within ST, particularly under weak coupling conditions [72]. This is because ST is an S-matrix theory, and the S-matrix seems not to make sense for asymptotically dS spacetimes [73, 74]. Turok countered this perspective, pointing out that our universe — and in particular cosmology — is not a scattering process, and asking whether ST could genuinely rule out the existence of a dS state. This exchange highlighted the ongoing debate over the compatibility of ST with cosmological models that incorporate a stable dS space, with panelists and participants expressing differing views on its implications, and once again pointing to the vastness of the string landscape and the need for predictive power. Basile and Van Riet noted again that also the SM has a vast landscape, so that the UV theory has to account for

that. Moreover, more impactful and solid conjectures often come with less predictive power. More generally, the statement is that the more rigorous a calculation is, the weaker the link to phenomenology. Vice versa, phenomenological models that can connect with data, have typically weak ties to the fundamental theory. The discussion then moved onto the topic of the non-perturbative existence of ST: While it is not proven that ST exists non-perturbatively, some evidence exists in some superselection sectors [55, 75–78]. The reason behind the difficulties in establishing the non-perturbative existence of ST is the fact that ST does not have a Lagrangian description at high energy, beyond the perturbative regime. This brought a final discussion on the relationship between predictivity, landscapes in QG, and the importance of continued dialogue and interdisciplinary collaboration in addressing the unresolved questions surrounding QG and its broader implications for theoretical physics.

In summary, while ST's computational and theoretical contributions were recognized, its current empirical shortcomings were debated. The panel agreed that ST addresses key issues in QG like non-renormalizability, but leaves questions about non-perturbative unitarity and predictivity open. There was broad acknowledgment of past overconfidence in ST's predictivity, but it was emphasized that current research is adopting a more cautious approach, with substantial progress having been made in the past decade. Disagreements emerged over the significance of the string landscape: Some saw it as a strength, indicating that "not everything (and almost nothing) goes in QG", others considered it a barrier to falsifiability, due to its vastness. It was nonetheless remarked that also the SM has a vast landscape of vacua, so that the landscape is built-in in any gravity-matter system. Ultimately, the need for interdisciplinary approaches and a focus on common QG principles was agreed upon.

## 2.5 Swampland: Yes or no?

> **Chair:** Gia Dvali
>
> **Panelists:** Ivano Basile, Alessandra Gnecchi, Renata Kallosh, Jan M. Pawlowski, Sumati Surya
>
> **Recording:** https://youtu.be/a_WElI00iu8

The panel focused on the significance of the swampland program across different approaches to QG. The discussion touched on different perspectives regarding the swampland's strengths and limitations, its relationship to BH physics, and its implications for cosmological models, including inflation and dark energy.

**Swampland: Stringy fever dream or universal framework?** The panel opened with the statement that the swampland program serves as a framework to delineate low-energy constraints on EFTs of gravity [65–69]. Dvali emphasized that the concept of the swampland includes established consistency conditions as well as the important notion of UV/IR mixing which in his opinion is a defining feature of QG. He pointed to the S-matrix as a crucial tool in understanding these connections (see also the related Panel 7). Basile advocated for a model-independent approach to the swampland, stressing that stringent principles in QG significantly narrow the range of viable theories. He suggested that consistency checks, such as BH thermodynamics and anomaly considerations, serve as rigorous filters in the swampland framework, and argued that nearly no EFTs satisfy these constraints. The core idea of the swampland program is indeed that of determining the set of theories admitting a consistent UV completion — not necessarily ST. Basile argued that the swampland is "model-independent", and can be used both to compare and to exclude theories. Gnecchi shared this viewpoint, adding that swampland constraints could be seen as a guiding principle for different QG approaches. The stringy side of the panel, as in the Panel 4, emphasized that almost nothing goes in QG, and that this is why the swampland idea is so powerful.

**Swampland, cosmology, and BHs.** Kallosh's insights primarily focused on cosmological implications, particularly concerning dark energy and inflation, in relation to the no dS swampland conjecture. She highlighted recent findings on quintessence models as alternatives to stable dS vacua [79,80], but also noted that empirical data strongly support a flat universe, questioning the validity of certain dynamical dark energy models. She further highlighted ongoing developments in ST-based inflationary models [81], underscoring that modular-invariant potentials could predict prolonged inflationary phases compatible with CMB observations [81]. While the no dS conjecture is debated even within the ST community, Gnecchi highlighted that swampland constraints based on EFT and BHs (e.g., the weak gravity conjecture (WGC) [82]) are much stronger and more established. She argued that examining the spectrum and microstates of BHs could provide insights into the transition from classical to semiclassical gravity, positioning BHs as a fundamental playground to understand QG and its IR constraints.

**Swampland beyond ST.** Surya introduced the idea of a "causal set swamplandia", emphasizing that each approach to QG might have its own swampland criteria. She discussed the unique properties of causal sets, such as scale separation, non-continuum phase, and the emergence of spacetime dimension — suggesting that the kinematic ontology of each theory affects its own swampland. She concluded that any valid theory of QG must be capable of reproducing standard cosmology and predicting consistent low-energy observables, such as a dS expansion and the spacetime dimensionality. Pawlowski added a comparative view by discussing overlaps between various QG landscapes, particularly those of ASQG and ST, based on an emerging effort in ASQG to delineate its landscape and compare it with swampland conjectures [83–85]. He highlighted that intersecting results across different QG approaches could point toward a more universally applicable "absolute swampland" [86].

**Additional discussions on a broader level.** The final portion of the discussion became technically dense, with exchanges on topics like anomaly matching, IR causality, and EFT constraints in QG. Dvali and Woodard debated whether certain formulations of pure QG might inherently reside in the swampland. Others discussed whether certain swampland constraints, such as the no global symmetries conjecture, might universally apply beyond ST: On the one hand, the stringy side of the panel argued that this conjecture stands on very solid grounds, and is independent of ST; on the other hand, Pawlowski pointed out that global symmetries may be compatible with ASQG. Regardless of the specific conjectures, Gnecchi suggested that different approaches should compare their predictions for BHs, as several swampland conjectures come from there. A lively, albeit occasionally fragmented, exchange unfolded around the no dS conjecture and whether current approaches adequately address QG's fundamental consistency requirements. In particular, Anselmi shared that criteria like unitarity should be taken seriously, but beyond that, it is unclear if other consistency conditions should be placed on an equal footing, see Panel 2.

In summary, the panel explored the role of the swampland program in QG. Several panelists agreed that the idea behind the swampland program could be extended to other approaches and guide research in QG. Although all participants did not share this, there was an emergent consensus that if the idea behind the swampland program applies beyond ST, then comparing landscapes [83,84] and pinpointing the *universal* criteria [83–87] that apply to all consistent theories remains an interesting challenge.

### 2.6 Discrete vs continuum approaches: Is spacetime discrete?

> **Chair:** Leonardo Modesto
>
> **Panelists:** Gia Dvali, Jerzy Lewandowski, Lesław Rachwał, Sumati Surya, Richard Woodard
>
> **Recording:** https://youtu.be/z3xDAaTAbZk

This panel discussion focussed on different interrelated questions about discreteness in QG, including the potentially discrete nature of spacetime itself or a mere discreteness of spectra of certain geometric operators, like the area operator in LQG [88]. Several arguments were made for and against discreteness.

**Discreteness yay, continuum nay.** Surya's citation "Be wise, discretize!" by Mark Kac, started the discussion of points in favor of discrete approaches. Several motivations for fundamental discreteness were put forward. It is well-established that any manifold can be represented by specific discretizations, like the one used in CST [89, 90]. Other motivations for discreteness include that spectra in QM are discrete (which might carry over to geometric operators as in LQG), that BHs likely have finite entropy and thus finitely many degrees of freedom, and that the discreteness could serve as a natural UV cutoff to regularize divergences. The latter can also be achieved in a Lorentz-invariant way, at least in some approaches like CST [91]. A continuum problem that, according to Surya, might be avoided in a discrete setting is the unenumerable sum over topologies. She also remarked that in CST discreteness is key to explain the smallness of the cosmological constant [92], and non-locality without the need for UV/IR mixing (since in CST there is a clear separation of scales) or causality violations. At the same time, Lewandowski emphasized that in discrete approaches re-obtaining diffeomorphism invariance is a priority. In LQG states are discrete, but not the observables, like for harmonic oscillators. He also emphasized how the introduction of holonomies is central in LQG, and each holonomy can be seen as a small amount of discreteness. Modesto interjected that, to him, introducing holonomies means to impose discreteness from the start, to which Lewandowski agreed.

**Discreteness nay, continuum yay.** So far, we can describe all observations with continuum QFT, and there is no experimental evidence for fundamental discreteness. An important question is whether discreteness solves any issues in QG, e.g., in high-energy scattering, and BH physics — a viewpoint put forth by Dvali. Any discrete approach also has to answer the question of how the continuum re-emerges, at least approximately. Lastly, as particularly emphasized by Woodard, a fixed discretization would not be consistent with observations, as inflation would have magnified any finite discreteness scale to a measurable length. Finally, the continuum limit is key for a theory to be predictive, namely, to have finitely many parameters. As opinioned by Rachwał, QFT is favored by an Ockham's razor argument: Continuum physics works, discrete physics is complicated.

**Non-renormalizability as a/the central topic in QG.** It was repeatedly emphasized by several participants that the non-renormalizability of GR is one, if not the, central problem, as also discussed in e.g. Panel 4. Different approaches have more or less direct resolutions. For example, ST is finite by construction, but one pays the price of a lot of extra structure. In ASQG, the issue is solved via an interacting renormalization group (RG) fixed point. Discrete approaches have a less direct answer, depending on the precise status that discreteness has. For example, in CST, discreteness is fundamental, and at least naively this takes care of standard UV divergences, but it is currently unclear how to uniquely fix the dynamics. In LQG, discreteness is first and foremost observed in the spectra of geometric operators, and the relation to the (non-)renormalizability is currently unclear.

**Hamiltonian evolution in discrete settings and "adding spacetime points".** A heated discussion ensued about whether or not spacetime points can be created or destroyed as time evolves. Woodard argued against such a creation, as the mechanism is unclear when fundamental degrees of freedom are viewed as fields and their derivatives specified at initial points. In CST, this seems to be no issue according to Surya. Dvali and Modesto also clarified that it is the Hamiltonian and the Hilbert space that are fixed, and the dynamics of spacetime points follows the unitary evolution implied by these.

**Causality and its violation.** Another point of discussion was whether discrete approaches necessitate causality violation. In particular, in CST every configuration itself is causal, but the effective causality originating from a superposition might be violated. This however needs a careful definition of causality [26], as also discussed in Panel 2.

**Bunch-Davies vacuum as a way forward?** A lot of attention in QG research is focused on areas without experimental data, like the information loss in BH evaporation. Woodard suggested that instead, an open issue for many QG approaches is to check whether they admit a Bunch-Davies vacuum to describe the cosmological evolution. A lot of data are available, e.g. for the power spectrum. To be consistent with the data, the existence of such a state is crucial. A counterpoint was made by Surya that model building only brings one so far, and that a solid understanding of the fundamentals is important.

Overall, no consensus was reached whether some form of discreteness is to play a role in QG, and a wide range of opinions was offered on how it would be realized: At a fundamental level, or at the level of spectra. Discrete approaches face unique challenges like recovering a macroscopic continuum description of spacetime (which is strongly connected to, and limited by, the available computing power for numerical simulations), but they also possess unique advantages like an intrinsic UV finiteness.

## 2.7 S-matrix approaches: Yes or no?

> **Chair:** Alessandra Gnecchi
>
> **Panelists:** N. Emil J. Bjerrum-Bohr, Mariana Carrillo González, Gia Dvali, Arkady Tseytlin, Richard Woodard
>
> **Recording:** https://youtu.be/thwXfQ3--L8

This panel discussed the role of the S-matrix in the formulation of QG, and also more generally in other theories. Various opinions were shared, ranging from "Do we really have to answer this question? Obviously yes!" to "In general, no!".

**Arguments in favor of using the S-matrix.** The S-matrix is one of the fundamental tools to formulate gauge interactions, including gravity. It sits at the heart of formulating high-energy physics observables like scattering amplitudes, and connects to theoretical consistency requirements like positivity bounds [93]. Importantly, as stressed by Gnecchi, the S-matrix is also key to identify signatures of new physics, such us in particle physics and GWs (see below). Bjerrum-Bohr added that the S-matrix is crucial to construct observables in QG and to connect the latter to EFT. The S-matrix (or more precisely, boundary correlators) is also central to the formulation of the AdS/CFT correspondence. There are stringent constraints on the form of the S-matrix, which are directly responsible for its predictive power.

**Limitations of the S-matrix.** There was universal agreement that there are aspects that the S-matrix does not describe. At a conceptual level, the S-matrix can only be defined if suitable asymptotic states exist, which might not be the case in all situations. A concrete example for this is cosmology, where however other observables (like correlation functions) are available. For practical computations, other formulations might be more suitable. For

example, as Tseytlin remarked, we do not understand confinement in QCD in terms of the S-matrix. Woodard criticized that the S-matrix is not really an observable in an operational sense, as asymptotic observers cannot compare their results in a causal way. He added that, at a practical level, we are not going to detect loop corrections to scattering amplitudes from QG — since the energy of the process is $GE^2 \sim 10^{-32}$; hence, the only possibility for QG to be constrained is via cosmology (see below).

**S-matrix and cosmology.** A major part of the panel discussion revolved around cosmology, and to what extent the S-matrix can be used to predict anything in this context. The shortcomings of the S-matrix were further sharpened: Cosmological particle production might preclude an asymptotically free vacuum at late times. In other words, within the S-matrix formulation (more specifically, in ST), the currently observed local dS phase cannot be a vacuum, and an eternal dS space cannot be an asymptotic state (see also the discussion in Panel 4). This point was particularly emphasized by Dvali: A cosmological background is not a vacuum. From this, it follows that the existence of the S-matrix would suggest that what we observe is not a constant cosmological constant, but rather quintessence. This is related to a viewpoint that emerged in Panel 4: The inexistence of a stable dS vacuum is a genuine prediction of ST, which is consistent with the impossibility of defining an S-matrix on it. A subtle related point is that the bare cosmological constant need not be zero, but the metric has to be asymptotically flat or anti-de Sitter (AdS) to admit an S-matrix.

**S-matrix and GWs.** There was general agreement that the S-matrix is very powerful for flat spacetime physics. This includes the description of GWs in the inspiral phase of BH mergers. Nevertheless, the S-matrix also seems to have limitations here. Carrillo González mentioned that tail effects in BH scattering might not be captured by the S-matrix — that is, the backreaction of gravitons on the background. It was also pointed out that in practical computations of waveforms, numerical relativity is an irreplaceable tool.

In summary, the S-matrix can be said to be a very useful tool when it is well-defined. Examples include high-energy scattering observables and the description of GWs. Nevertheless, it has shortcomings: In some situations, it simply lacks a definition (like on dS space), and there was universal agreement that the S-matrix is not the be-all and end-all.

## 2.8 Perturbative vs non-perturbative approaches: Is quantum gravity non-perturbative?

> **Chair:** Richard Woodard
> **Panelists:** Leonardo Modesto, Jan M. Pawlowski, Lesław Rachwał, Arkady Tseytlin
> **Recording:** https://youtu.be/OcA8oeosPbc

This panel discussion revolved around questions on what aspects of QG are really non-perturbative, what this even means, and how one could compute non-perturbative effects.

**What are non-perturbative effects?** Two different kinds of "non-perturbative" phenomena were discussed during the panel. The first kind falls under the label all-order/resummed perturbative effects. Many phenomena can be understood within this category, but some important effects cannot be captured within this framework. The latter could be called "truly non-perturbative". This category includes instantons (like the Eguchi–Hanson spacetimes), spacetime topology change, or collective phenomena. Specifically, Rachwał pointed out that in perturbative QG, we cannot see topology change, not even resumming all-order effects. For instance, if one considers an instanton, the transition amplitude goes like $e^{-1/g^2}$, which cannot be expanded around small $g$, and hence cannot be described by any perturbative resummation. Moreover, we do know that non-perturbative (non-linear) effects are already needed at

the level of GR, e.g. to describe a BH merger, and in the low-energy limit of QCD. These motivations kicked off the discussion.

**Non-perturbative: With respect to what?** What we usually call perturbative refers to an expansion about the trivial saddle point. In this sense, what is perturbative about one saddle could appear non-perturbative about a different one. The choice of expansion point is then crucial in computations. This is directly connected to the idea of resurgence [94].

**Non-perturbativity in non-gravitational theories.** Some examples of non-perturbative effects in non-gravitational theories were discussed, from which we might hope to learn more on what to expect in QG. Pawlowski pointed out that it is not always easy to disentangle perturbative from non-perturbative effects: The Schwinger effect (the production of electron-positron-pairs in a strong electric field) can be computed by resumming one-loop Feynman diagrams, but it is inherently non-perturbative. It was also discussed whether reaching a strong coupling regime automatically signals that new degrees of freedom have to be introduced. This point was put forth and supported by Dvali, who gave the case of QCD as an example: When pions are strongly interacting, quarks must appear. Pawlowski disagreed on this point, since in principle strong coupling physics can be described in terms of the same degrees of freedom: One can describe QCD in the IR in terms of quarks and gluons, it is just ill-advised. Overall, no agreement about this point was reached.

**Non-perturbative tools.** Some useful tools that were mentioned during the discussion and that are currently used include integrability, the AdS/CFT correspondence (which Tseytlin noted could be used to make progress in QG independently of ST), and diagrammatic approaches like the functional RG. Nonetheless, there was a general consensus that (new) non-perturbative methods are essential to make more progress in QG.

**Defining a graviton state.** There was an intense debate about the definition of a proper graviton state. In certain regimes, a linear graviton might be a good effective description, in the same sense as a gluon is at high energies in QCD. However, due to the non-linear symmetry of gravity, the true graviton state must be much more complicated, in particular at high energies where the theory is potentially non-perturbative. This issue ties in with the discussion of what happens in high-energy graviton scattering, and a range of opinions was provided in the panel, ranging from "clearly, high-energy scattering produces BHs" to "what does it even mean to scatter individual linear gravitons at high energies?". This topic also appeared in Panel 3.

**The scale of QG.** Another item of discussion was the scale of new or non-perturbative physics in QG. In ST the string scale must be below the Planck scale in order to achieve weak coupling. A relevant quantity in this context is the species scale. In EFT, if we consider $N_{sp}$ species interacting with gravity, then the cutoff can be lower than Planck mass, specifically it can be $M_{Pl}/\sqrt{N_{sp}}$ [95,96]. For instance, in the case of the SM we have roughly $N_{sp} \sim 100$, so the cutoff is lowered by a factor of $1/10$. By contrast, in ASQG and other theories, the scale is believed to be the Planck scale. Due to the arguments related to the species scale, this also means that theories where QG lives above the Planck scale are inherently non-perturbative. Related to this point, Dvali also advocated that if QG would be above the Planck scale, then because of classicalization effects, we practically would not need it: High-energy scattering produces BHs beyond the Planck scale, preventing us from probing even higher energies.

Overall, there was a consensus that non-perturbative effects can play an important role in QG, be it in the form of gravitational instantons or the proper definition of a graviton state. While difficult, it could be essential to understand the fully non-perturbative regime in QG and to answer fundamental questions about, e.g., BHs or topology change.

## 2.9   Quantum cosmology

> **Chair:** Guilherme Franzmann
>
> **Panelists:** Robert Brandenberger, Anne-Christine Davis, Paulo Moniz, Alberto Salvio, Kellogg Stelle
>
> **Recording:** https://youtu.be/bvF9zW9j028

The discussion was initiated by Franzmann on fundamental questions about the physical understanding of the wave function of the universe, the emergence of spacetime, primordial physics and its connection to the quantum nature of gravity, and the relation between QG, dark matter and dark energy. Furthermore, fundamental aspects of quantum field theory on curved spacetime (QFTCS), the violation of unitarity, and a possible breakdown of the EFT description of gravity in the cosmological evolution were highlighted.

**Universe as a quantum object and path integrals.** Salvio noted that it is impossible to describe the universe without accounting for quantum physics, especially in the context of the early universe. He pointed out that observables require observers, so only in those patches of the universe that are suitable for life (and so for observers) we must require observables to have all the standard quantum-mechanical properties. In this regard, Moniz mentioned the unsettled issues of the quantum-to-classical transition, and of deriving classical observables. The panel also discussed the path integral framework for the wave function of the universe [97]. Stelle highlighted that some of the latest results on Lorentzian path integrals [98, 99] are consistent with the Euclidean approach in the late time evolution of the (quantum) universe. However, there was also a debate on the validity of the Euclidean path integral in gravity and the Wick rotation of the results to Lorentzian signature (see also Panel 1 and Panel 4).

**Problem of initial conditions.** Moniz highlighted the emergence of quantum chaos from classical non-linear dynamics due to the fact that very close but unequal initial conditions give rise to completely different late-time cosmologies. He emphasized that the failure of the Wentzel–Kramers–Brillouin (WKB) method in QC in determining the wave function of the universe is an important roadblock that needs to be cleared in the future [100]. A potential way out was mentioned by Stelle: He mentioned some recent results in higher-derivative gravity [98], indicating that with the finite Euclidean action principle [101], one can suppress the anisotropic cosmologies that underlie the above issues and have "a safe beginning of the universe" [98]. This suppression would lead to a nearly-isotropic universe, which is in agreement with observations. In particular, recent results in quadratic gravity might suggest that Starobinsky inflation [52] could be compatible with asymptotic freedom [102].

**Dark matter and dark energy.** Davis highlighted that we do not know what 95% of the universe is made of. Yet, there is a big difference between dark matter and dark energy. The first indicates that there is much more in galaxies that we have not discovered yet at the Large Hadron Collider (LHC). The second is, in her opinion, much more mysterious, and its resolution could lie in the foundation of how QM and gravity combine. Some of the panel and participants expressed skepticism on recent results of DESI [103], which suggest dynamical dark energy. Davis suspected that the problem of the Hubble tension in cosmology might have a fundamental resolution in astrophysics rather than cosmology. The panel discussion has acknowledged dark matter as the most perplexing problem in cosmology. Some participants wondered if the acceleration parameter $a_0$ in Modified Newtonian Dynamics (MOND) [104] has any fundamental role in enriching our understanding of dark matter. Further discussion has brought attention to the peculiar nature of dark matter in shaping galaxy clusters and the formation of halos, and this led to the question of whether dark matter should be non-minimally coupled to gravity. Everyone agreed that the connection between QG and dark matter could only make sense if primordial BHs acts as dark matter.

**Applicability of EFT in cosmology.** Brandenberger made the point that the EFT paradigm may be fundamentally inapplicable in cosmology [105], particularly in the early universe. This stems from two key principles: The trans-Planckian censorship conjecture (TCC) [106], and the breakdown of analyticity and causality in graviton scattering at the species scale. Spacetime itself may be an emergent phenomenon, and this should manifest in observable signatures, such as GW and cosmological perturbations. In particular, he noted that string gas cosmology and some matrix models predict a blue tilted primordial GW spectrum [107]. Brandenberger also stressed that QG is essential for determining the initial conditions at the onset of the standard Big Bang epoch. However, its role appears to be disconnected from the problem of dark matter, as also pointed out by Davis. At variance with Davis though, he opinioned that the cosmological constant problem is not a genuine issue; it arises from applying EFT in a regime where it is not valid. At the same time, dark energy is central to understanding QG. It is neither a cosmological constant nor quintessence, and must be explained without fine-tuning. According to Brandenberger, the theory that successfully achieves this will likely be the correct one. Notably, the paradigm of standard slow-roll inflation [108] is no longer viable under the constraints of the TCC [109]. This also implies that dark energy cannot be interpreted as a simple cosmological constant. In essence, Brandenberger summarized his thoughts by voicing that inflation should finally "rest in peace".

**Additional discussions on a broader level.** In the context of bouncing cosmologies, as Brandenberger pointed out, the observable scales today do not originate from a trans-Planckian regime, thus bypassing the TCC. This raised the question whether inflation is worse than bouncing models. Dark matter remains a significant challenge, with both Davis and Brandenberger agreeing that it cannot be explained by QG. Additionally, Liberati shared that he noted a growing skepticism within the astrophysical community regarding the assumption of cold dark matter. Regarding dark energy, emergent gravity theories suggest that the cosmological constant must be computed from a fundamental theory. Franzmann raised a question regarding the emergence of time, citing the potential transition from a Euclidean to a Minkowski signature. Wick rotation — a crucial aspect of this transition — does not always lead to a well-defined Lorentzian theory [110]. Regarding observational data, Davis emphasized the great precision of Planck satellite measurements. Despite this, both Davis and Sakellariadou expressed growing doubts about the Lambda cold dark matter (ΛCDM) model. While ΛCDM is still remarkable, Sakellariadou noted that its success relies heavily on its use as a prior, and cracks in the model are beginning to show. As for future observations, Brandenberger highlighted that the tensor tilt could provide crucial insights into QG, potentially distinguishing between competing cosmological models. To this, Salvio added that a peculiar prediction of the CMB power spectrum from a specific QG theory could serve as a test for QG. Liberati closed the discussion by touching again on MOND, acknowledging its success in fitting galactic rotation curves with just one parameter but noting its failure in explaining galaxy clusters.

In summary, this panel discussed the aspects of the universe as a quantum object and pointed out the common features and differences between Lorentzian and Euclidean path integral approaches in QC. Questions were raised on the physical interpretation of the results obtained using the Euclidean approach. The panel compared the initial conditions of the universe within GR and higher-derivative gravity. There were discussions on the possible breakdown of EFT because of the TCC and its implications for early and late-time cosmology in ruling out inflation and the cosmological constant. Although no agreement was reached on this, there was consensus on the importance of understanding both physics beyond ΛCDM and current observational uncertainties. Finally, everyone agreed that QG is essential in deciphering the nature of dark energy.

## 2.10 Probing quantum gravity with future observations

> **Chair:** Stefano Liberati
>
> **Panelists:** Anne-Christine Davis, Giulia Gubitosi, Paulo Moniz, Mairi Sakellariadou
>
> **Recording:** https://youtu.be/girDk3Dej9Y

This panel discussed a number of possible ways to test QG: Different theories with current and upcoming astrophysical and cosmological probes, including GWs, and the quantum nature of gravity via tabletop experiments. Liberati initiated the discussion by raising the question of what kind of new physics could be induced by QG, in particular he mentioned Lorentz violation and non-locality. He also stressed the importance of being able to probe QG regimes without the need to use single-particle probes of Planckian energy, and that astrophysical processes could be useful in this respect [111].

**Signatures of Lorentz violation and QG.** Lorentz invariance could be broken in some QG approaches and induce modifications in the dispersion relations. In this context, it was remarked that there are both tight observational constraints on Lorentz violation from astrophysical observations [112], but also many observational uncertainties involved [113]. There was also a debate on whether Lorentz-violating QG approaches are compatible with renormalizability: Liberati mentioned Hořava gravity [114] as an example, since it is Lorentz-violating and renormalizable, at least in the projective case, but likely does not resolve singularities. There was a debate on whether and how any signatures of Lorentz violation can actually pinpoint a specific approach to QG, and whether Lorentz-violating approaches are solidly ruled out. For example, Knorr pointed out that the RG flow in such systems tends not to restore Lorentz symmetry at low energies [115, 116].

**Detecting modified gravity and connection to QG.** Platania raised the question of whether there is any experiment that, by detecting deviations from GR, could conclusively relate it to QG effects and not some classical modification of GR. Sakellariadou and Davis expressed caution about detecting any signatures of modified gravity and connecting them with a specific theory of QG. They urged the QG community to look for a fundamental derivation of gravity modifications. In particular, Sakellariadou emphasized that dark energy phenomenological models, even if successful, must be explored further until the models are understood at the fundamental level. GWs were agreed to be a promising way to probe new physics beyond GR, including new degrees of freedom and changes in the spacetime dimension. The panel agreed that so far, there is no evidence in this direction. Liberati highlighted the severe constraints on extra-dimensional gravitational theories from the observation of binary neutron star mergers. In this regard, Gubitosi voiced that we need to understand QG in high-density environments, while Davis remarked that detecting modified gravity cannot confirm or rule out the presence of extra dimensions, unless their meaning is fundamentally understood in QG. Moniz claimed that the detection of torsion in the CMB or in astrophysical tests would be a strong indication in support of supergravity, but this was not shared by all participants. Liberati noted that the constraints on torsion from binary pulsars have to be considered in building models of the early universe cosmology. It was also discussed whether resolving BH singularities in QG can result in any observational signatures at the horizon scale, that could be potentially detected through BH shadow and GW experiments (see also Panel 12). Sakellariadou pointed out the need to take into account observational uncertainties in this regard.

**Quantum nature of gravity.** Buoninfante pointed out that it is important to distinguish between testing QG approaches and detecting the quantum nature of gravity. In the latter context, two setups primarily emerged.

(i) **Tabletop experiments.** These experiments aim at testing the quantum nature of gravity in a laboratory, i.e. in low-energy regimes [117–119]. Gubitosi mentioned that impressive technology is being developed to place a massive particle in a quantum superposition. For example, achieving quantum superpositions for masses of order $10^{-14}$ kg could in principle allow to detect gravity-induced entanglement between two free-falling superposed massive particles [119]. However, realizing such an experiment still requires some time, since the largest massive particle ever placed in a quantum superposition has a mass of the order of $10^{-20}$-$10^{-21}$ kg. Davis pointed out other laboratory experiments that could instead be useful to test modified gravity such as the torsion balance [120], the Casimir force [121], and the atom interferometers [122]. This further elevates the importance of tabletop tests for gravitational quantum physics and physics beyond GR.

(ii) **Primordial GWs and the CMB.** There was a question on whether detecting primordial GWs would indirectly validate the quantum nature of gravity, on which some of the panel and participants have expressed an opinion that the answer is no. It was also pointed out that GWs could come from other sources like cosmic strings or a bouncing cosmology, and by observing them one cannot decisively determine the underlying physics. Gubitosi remarked that the signatures of parity-violation in the CMB (that were claimed to be found recently [123]) are worth paying attention to. Moniz mentioned that detecting quantum-gravitational corrections from the Wheeler-DeWitt equation to the CMB low-$\ell$ angular power spectra [124] is another interesting arena, but he also acknowledged the difficulties with huge uncertainties due to cosmic variance. Liberati commented on the observed flat homogeneous and isotropic geometry that could result from a sum over several topologies, for which Sakellariadou responded with skepticism on finding them in the CMB.

In summary, current and upcoming experiments aimed at testing deviations from GR and quantum gravitational effects were inspected. It was highlighted that cosmological and astrophysical probes, such as GWs, are promising ways to test deviations from GR. These, however, might not give us direct evidence of the quantum nature of the gravitational interaction. Furthermore, it was pointed out that progress has been made towards the possibility of realizing tabletop experiments aimed at detecting the quantum nature of gravity in low-energy regimes. However, there is still a long way to go.

## 2.11 Quadratic gravity, effective field theory, and modified gravity: Use and limitations

> **Chair:** Alberto Salvio
>
> **Panelists:** Robert Brandenberger, Astrid Eichhorn, Giulia Gubitosi, Lavinia Heisenberg, Kellogg Stelle
>
> **Recording:** https://youtu.be/6ErXUvX6gQA

To study quantum aspects of gravitational interaction and make contact with experiments, in principle, we might not need a fully non-perturbative and complete understanding of the quantum-gravitational interaction (some of the panelists of Panel 8 might want to disagree). Just as we can study GWs as small perturbations at the classical level, we can hope to capture some quantum features of gravity by using a perturbative approach or by considering only a few local terms in the QG Lagrangian. Salvio opened the discussion by introducing (i) the EFT of GR and (ii) a perturbatively renormalizable UV completion known as quadratic gravity.

(i) The EFT of GR [125, 126] consists in adding all possible terms to the Lagrangian that are compatible with the symmetries, e.g., with diffeomorphism invariance. In this case, the massless graviton is the *only* gravitational degree of freedom, and the cutoff scale is expected to be the Planck mass, which can be lowered if matter species are coupled to gravity [95, 96, 127]. The EFT framework provides powerful QFT calculational tools and allows to make predictions at energy scales lower than the cutoff, e.g. quantum corrections to the Newtonian potential can be consistently computed [128].

(ii) If additional particles are allowed in the gravitational spectrum, in particular a massive spin-zero coming from the square of the Ricci scalar, and a massive ghost-like spin-two coming from the square of the Weyl tensor, one has a gravitational theory (quadratic gravity) that is perturbatively renormalizable in four spacetime dimensions [3–5, 37, 38, 102]. In this case, the bare Lagrangian contains a finite number of terms.

The panel discussion focused on the regime of validity of these approaches, whether they can be useful to explain new physics beyond GR in the early universe cosmology and in BH physics, specifically if EFT and/or quadratic gravity are able to tell us something about BH horizons and singularities.

**Use and limitations in cosmology.** Various disagreements arose, especially in relation to early universe cosmology. Although the EFT description (without the addition of ad hoc matter fields) is not able to describe the anisotropies that we see in the CMB radiation, some of the participants agreed that the renormalizable theory of quadratic gravity does instead provide the best explanation to date of the inflationary era. This is due to the presence of a suitable spin-zero degree of freedom that plays the role of the inflaton (Starobinsky inflation [52–54]). On the other hand, another group of participants criticized this point of view, and claimed that any local and perturbative description of the early universe would break down at energy scales lower than the inflationary one. One of these claims is supported by the TCC [105, 129], as emphasized by Brandenberger. Platania noted however that the conclusions of the TCC can be avoided in UV-complete non-perturbative QFTs of gravity [34], such as ASQG.

**Use and limitations in BHs.** Recent work has shown that BHs with singularities are still solutions of the classical field equations of quadratic gravity [130, 131]. Part of the discussion highlighted that the problem of curvature singularities could be solved by taking into account new gravitational terms in the action. Eichhorn and Platania remarked that these additional terms could arise as the result of a fully non-perturbative approach [132–134]; Knorr mentioned that higher-curvature terms of at least cubic order could be already sufficient to tackle the singularity problem [135–137]; other participants observed that the presence of non-local form factors could also play a key role in obtaining regular spacetime solutions [138, 139].

**Additional discussions on a broader level.** Part of the panel also focused on whether quadratic gravity could have a non-perturbative completion. In particular, Stelle proposed that ASQG could provide a new understanding of the spin-two ghost at the non-perturbative level [140]. Furthermore, it was also questioned whether (local) Lorentz invariance is fundamental at the microscopic level. For example, Liberati argued that if spacetime is quantum, then it could be characterized by an atomistic (discrete) nature that breaks (local) Lorentz invariance. In this case, one hopes that the problem of singularities could be more easily solved thanks to the discretization of spacetime (see also Panel 6). However, it was also pointed out that this kind of approach would suffer from the same problems as all discrete approaches to QG, namely the lack of understanding of how to recover a continuum description at low energies and large distances.

In summary, this panel discussion highlighted the lack of consensus on the role of the EFT framework and the higher-curvature terms in explaining the early universe evolution. A group of participants agreed that the renormalizable theory of quadratic gravity can explain

the inflationary era and the formation of CMB anisotropies without introducing ad hoc scalar fields. On the contrary, another group of participants proposed that any local perturbative approach would fail at the scale of inflation and that a non-perturbative mechanism or some form of non-locality is needed. Furthermore, different ideas were shared about the role of the higher curvature terms for the resolution of BH singularities. While local quadratic curvature terms still allow for BH singularities, the presence of cubic and/or higher order curvature terms, including non-local ones, could help solving the singularity problem.

## 2.12 Black holes in quantum gravity

> **Chair:** Kellogg Stelle
>
> **Panelists:** Astrid Eichhorn, Stefano Liberati, Olga Papadoulaki, Mairi Sakellariadou, Francesca Vidotto
>
> **Recording:** https://youtu.be/u0KWQcnUL0U

This panel focused on BHs in QG and specifically on their role in connecting and constraining UV physics. Stelle opened this discussion session with a stimulative list of topics and questions such as gravitational corrections at BH horizons, non-Schwarzschild solutions in QG and their stability, ultracompact horizonless objects, extremal BHs and swampland conjectures, BH remnants, wormholes, dynamical formation and evaporation of BHs, and the information loss paradox. The discussion then revolved around a broad range of topics, from theory-agnostic considerations on BHs beyond GR to specific models and observational windows to detect deviations from classical GR.

**Astrophysical BH and QG effects.** The discussion was kicked off with the question of whether quantum-gravitational effects could be observed in astrophysical BHs. Eichhorn opinioned that this is unlikely if the scale of QG is around the Planck mass and if BHs are slowly spinning. However, high-spin BHs might present a window into QG [141]: If quantum BHs are regular, there is a chance that the two horizons are absent, resulting in a direct view into the quantum-gravitational region, and in additional light rings in the images of BH shadows [142]. While GR makes it difficult to dynamically reach the critical spin required for this scenario, QG could alter this conclusion. Acknowledging this point, other panelists pointed out that one could look for deviations from GR by looking at the structure of photon rings whose apparent location is mostly independent of astrophysical details [143].

**Observational prospects and challenges.** Sakellariadou turned the discussion towards observational signatures of quantum-gravitational phenomena. She voiced that strong lensing [144], gravitational waveforms (including echoes), and quasi-normal modes are the most promising arenas to look for QG signatures [113, 145, 146], but noted the significant astrophysical uncertainties involved. The question remains: Which approaches to QG offer the best hope for discovery? It was pointed out that answering this question is very difficult at the moment: There are either precise predictions that do not belong to the observational window, or QG-inspired models whose connection with QG is weak. Some panelists in particular expressed skepticism about the observational prospects for quantum-gravitational effects, arguing that classical collapse in GR offers little room for detection after an event horizon has formed.

**Structure of quantum BHs and stability.** A consistent theory of QG should avoid singularities, and thus BHs must have some form of regularity. Liberati pointed out some of the structural possibilities based on purely geometric considerations [147], but then focused on two of them: Those with outer and inner horizons (i.e., regular BHs) and those with a minimal radius, resembling wormholes. The first category might be subject to two types of instabili-

ties at the inner horizon. The first is mass inflation [148–150] — a classical instability that arises from the exponential blueshift of infalling matter. The second is a semi-classical instability [151], which is due to an incompatibility between a regular Unruh vacuum state at the inner horizon and one at the outer one. These instabilities might be mitigated in BHs characterized by a vanishing surface gravity at the inner horizon [152], but this is a highly fine-tuned solution. Extremal BHs might be free from such instabilities in principle, but Liberati warned of the risks associated with the formation of inner light rings, which occur when horizons merge. These structures could exacerbate instabilities unless mitigated by specific dynamic solutions. Some panel members suggested that ultracompact horizonless objects (or BH mimickers) could be an interesting alternative to BHs. Among such objects, Stelle mentioned that fuzzballs are a concrete singularity-free alternative, which may also provide a solution to the BH information paradox. Vidotto questioned the need to look for BH mimickers and commented in this regard that it is more important to understand BH interiors, since these cannot be treated with classical physics.

**BHs in LQG.** Vidotto provided an LQG perspective on the discussion, asserting that real BHs are regular and paradoxes arise from assuming that singularities are there. She pointed out that QG effects at the horizon rely on the details of the BH interior, and mentioned certain developments in LQG including details of dynamical collapse, BH to white hole transitions, and BHs formed by a non-singular collapse due to the existence of a minimal length scale [153], and in analogy with other approaches to QG. Vidotto also touched on the question of BH entropy, pointing out that different frameworks define entropy differently; this contributes to the confusion around concepts like the information loss paradox and holography. She also warned against treating holography as an unquestionable principle.

**BHs in ST.** Papadoulaki remarked that it is very important to understand microscopic models underlying BH physics. She emphasized the success of ST in understanding supersymmetric BHs, particularly through the explicit microstate counting [70, 71]. She highlighted that holography and the AdS/CFT correspondence are powerful non-perturbative tools for studying BH [154], but acknowledged their limitations for non-supersymmetric cases. She noted the success of ST in preserving the unitary evolution of BHs with asymptotically AdS boundaries [154], but it was commented that a quantum-mechanical understanding of realistic asymptotically Minkowski BHs is still work in progress. The use of lower-dimensional models, such as Jackiw-Teitelboim gravity [155] and the $c = 1$ Liouville ST [156], simplifies microstate counting and provides insights into macroscopic entropy [157]. The later discussion questioned the status of BHs in ST and whether or not ultracompact objects like fuzzballs [158, 159] are robustly understood. The panel noted a lack of consensus in the ST community about the fuzzball paradigm. Furthermore, it was noted by Sakellariadou that there are no precise predictions derived from the fuzzball paradigm to test them robustly with observations.

**Theoretical perspectives on BH evolution.** An interesting question that emerged in the discussion is whether all the static BH solutions we find in theories of QG can be considered physical, or whether their dynamical formation process plays a role. Some of the panelists and participants acknowledged that not all the BH solutions can be realized in nature, and to understand what are the viable ones, it is important to study their dynamical formation. As an example, Stelle mentioned that quadratic gravity admits a class of non-Schwarzschild BH solutions [130], but noted that their physical significance is still an open question. The panel also debated the role of entropy in BH evolution. Buoninfante and Liberati proposed that the BH collapse might involve a phase transition from volumetric entropy to area entropy, with the latter dominating as compactness increases. Alternatively, during collapse, there might not be a clear discontinuity between matter and gravitational contributions, but rather a gradual transition where the latter dominates over the former. Related to the question of evaporation

and unitarity, there was also some discussion about the no global symmetries conjecture, and Eichhorn commented that its status in ASQG is unsettled [83, 85, 86, 160] (see also Panel 4).

**Remnants as dark matter candidates?** Some participants asked whether or not BH remnants can be dark matter candidates, or if they can be ruled out by astrophysical and cosmological observations. Sakellariadou commented that there are multiple layers of astrophysical uncertainties in this context, and it is non-trivial to place any constraints on BH remnants as dark matter [161, 162]. Vidotto remarked that to understand BH remnants, one must investigate BH interior geometries from a QG perspective. In this context, she made a comparison between the string length scale and the minimal length in LQG, and claimed that they share some features in understanding the interior geometry of BH remnants [153, 163].

In summary, the panel touched on a wide range of topics, from theoretical modeling to observational challenges, offering diverse perspectives on BHs in QG. Some theory-agnostic considerations have been made about the type of BH alternatives and their features. Despite the richness of the topics, the panel highlighted that there is a long way to go in deriving predictions on quantum BHs, and in identifying their observational signatures. Nevertheless, the session underscored the progress made and the open questions that remain in understanding BHs within QG, including their formation and the dynamical evolution of their entropy.

# 3 Individual thoughts in the light of discussions:
## *Formal aspects and consistency of quantum gravity approaches*

### 3.1 Damiano Anselmi:
#### *Perspectives in quantum gravity*

Thanks to the unprecedented achievements of the SM, QFT has emerged as the most successful framework for high-energy physics and arguably the best candidate framework for QG. In recent years, there have been significant advances in higher-derivative and non-local QFTs of gravity, helping us better understand various issues and providing new calculational techniques. Many of these have been discussed in Panel 2 and Panel 3. Here we mention two methods for removing ghost particles: One is to turn them into purely virtual particles (particles that are never on the mass-shell) within the scope of local QFT [4, 164–167], and the other is to cancel the unwanted poles of propagators by means of non-local deformations [45, 46, 168–176]. A typical trade-off for achieving both unitarity and renormalizability in quantum gravity is the violation of microcausality, which remains consistent with experimental data if the energy scale of violation is sufficiently high.

Non-local theories are intriguing in their own right, despite their inherent arbitrariness. They may be considered fundamental theories or as tools to investigate new aspects of local QFT. However, the relationship between local and non-local theories is not yet fully understood. I would like to emphasize that this is an important area of research that could enhance our understanding of both.

Among the many strengths of QFT is the ability to formulate it off-shell [177], using Lagrangians or path integrals to represent all possible configurations. In contrast, other approaches, notably ST, encounter significant technical challenges in this regard. These include the lack of a string Lagrangian, the absence of a systematic method for extending ST beyond the standard worldsheet path integral to an off-shell formulation, and the lack of a complete string field theory.

A framework constrained by such limitations is unsatisfactory, as it cannot reliably provide candidates for fundamental physics. Meanwhile, the ST community often focuses considerable effort on peripheral issues that appear minor by comparison. One of these, discussed in Panel 2,

is the landscape/swampland consistency criteria. This set of conjectures and guidelines aims to distinguish EFTs that can arise from a theory of QG from those that cannot. Some well-known swampland conjectures include the swampland distance conjecture, the WGC, the no dS conjecture, the no global symmetries conjecture, and the cobordism conjecture. Moreover, aspects of the swampland program impose constraints based on causality. The underlying idea is that many theories, which appear well-behaved under QFT, may break down in the presence of gravity or when attempting to incorporate them into a more fundamental framework.

While ST is often regarded as such a foundational framework, this view is challenged by its inability to go off-shell, as previously discussed. Furthermore, the conjectures mentioned seem overly lenient toward ST itself. For example, no requirement highlights the necessity of going off-shell (a crucial aspect for accurately describing the physical world), which would exclude ST from consideration. Additionally, causality-based criteria lack solid grounding, as there is no physical reason for causality to be fundamental, or a criterion for selecting consistent theories.

In my opinion, ST pales in comparison to QFT. Ultimately, we must question the rationale behind investing time and resources in swampland consistency requirements when ST faces much larger issues at its core. Going forward, I hope the community will recognize these limitations and reconsider the value of continued investment in this direction.

## 3.2 Ivano Basile:
### *Unraveling Ariadne's thread*

**Finding common ground.** One of the main key takeaways from the panel sessions, in particular Panel 1 as well as Panel 5, is the role played by (gravitational) EFT. As a solid, cohesive and empirically successful framework which systematically incorporates ignorance about new physics, it ought to act as the common ground for QG approaches. To some extent, this sentiment appears not to be shared by the entire community. In order to make concrete progress in connecting the communities, it is of crucial importance to settle on a common ground to start building on — in other words, an agreement on the relevant questions is necessary to compare answers and evaluate proposals. To this end, gravitational EFT is the natural starting point to start unravelling the metaphorical thread of physical reasoning.

**Swampland ideas as a guide.** In this spirit, the bottom-up model-independent developments in the context of the swampland program offer a number of hints on relevant directions to pursue. The nature of BH entropy and observables in QG are at the core of this web of ideas, and discussing the various perspectives on these matters in a constructive and detailed fashion is paramount. A key takeaway in this respect is that the consistency with the basic physical principles of unitarity, causality and (possibly) stability is significantly more constraining than in non-gravitational QFT. A thorough comparison of the implications of the swampland program is likely to shed light on the consistency of any given proposal for a theory of QG.

**Directions in ST.** Thus far, the top-down applications of this type of reasoning have been mostly (though not exclusively [83, 84, 86, 87, 160]) confined to ST. While healthy inter-communal discourse would surely benefit from a broader investigation along these lines, the discussions in the workshop also pointed to well-known desiderata in this particular field, namely obtaining predictions which are as sharp and as experimentally accessible as possible. While the former is a theoretical issue, the latter is a phenomenological one — the high-energy signatures of string scattering have been studied in several works dating back to [178], while low-energy features of any given string vacuum are in principle calculable as those of any given QFT. As such, there is no methodological difference between the frameworks at this level. ST features many sectors and vacua, as expected by any theory of QG (including the SM EFT [179]), which generically entails that low-energy predictions bring along a trade-off between sharpness and genericity. However, recent results may point to a way out of this

unpleasant state of affairs [180–182]: The precise low-energy consequences of ST, at least in some (e.g. weakly coupled) sectors, may be more universal than previously appreciated. Among these possibilities is the emergence of mesoscopic dimensions [181], which *a priori* is not required since non-geometric sectors are present in the string landscape.

**Observables and holography.** The present big-picture vision for future collaborative endeavors across the field hinges on understanding observables and the related role of the holographic principle, embodied by the S-matrix or more generally boundary observables. Following the proverbial *Ariadne's thread* of EFT, there are reasons to believe that the only (nonperturbatively) sharp observables are defined on an asymptotic boundary, which among other things would exclude eternal dS from the superselection sectors of QG. Together with other semiclassical considerations, one is led to the holographic principle, whose ramifications in connecting different proposals for QG would be very instructive to explore further. All in all, the general philosophy outlined in this vision advocates to begin from the common ground of EFT, take its lessons seriously and unravel its surprisingly far-reaching implications for QG.

### 3.3 N. Emil J. Bjerrum-Bohr:
*Quantum gravity and general relativity from amplitudes*

A modern strategy for perturbatively quantizing gravity [183,184] starts by extending the Einstein-Hilbert action with EFT couplings [185]. Not only does this permit a clear and universal scale separation of low-energy long-range physics [125,128,186,187] from UV short-distance effects [188], it also offers an exciting alternative to the classical geometrical framework of GR that can incorporate various gravitational phenomenology. As a weakly coupled QFT mediated by quanta — massless spin-two gravitons — gravity compares to traditional perturbative expansions of electro- and chromodynamics, and while conventional off-shell perturbative QFT methods [189,190] tend to be unwieldy, recent advancements have shown how to bypass excessive complexity in calculation [191,192]. For instance, utilizing on-shell unitarity-based computational techniques — initially designed for SM physics — one can cleverly leverage the incredible Kawai-Lewellen-Tye gauge-gravity relations [193–195] to compute graviton scattering of BHs [196–198] — taken as point-like particles! This versatile laboratory of computational techniques provides a flexible and efficient framework based on QFT that is increasingly becoming essential for rigorously testing Einstein's theory of gravity from observations, for instance, [199,200].

Current research explorations focus on providing accuracy to theoretical expectations for observables as an alternative to classical numerical GR. In these applications, classical physics emergence is assured by quantum mechanics' fundamental correspondence principle. A post-Minkowskian expansion is usually preferred [201,202] because it ensures correct classical results even at relativistic velocities. One can also investigate the consequences of phenomenological extensions of gravitational interactions by adding additional couplings to the formalism or focus on perturbative computations of waveforms. While a major priority has been computations for BHs with no classical spin, significant progress has been accomplished for spinning BH observables [203–207]. The development of a new bootstrap method has enabled the expression of tree-level classical spin amplitudes in terms of entire functions free of spurious poles [208,209]. This recent progress in spinning BH computations offers exciting theoretical prospects for an all-order spin QFT framework. While considering subtle IR and radiation cancellations, efficient loop integration strategies combined with compact and practical integrand generation are crucial for progress [210].

Although tiny and currently outside the reach of observations, the EFT treatment of GR allows us to explore the consistency of quantum terms in scattering amplitudes [211, 212] and their potential consequences for observables. Another exciting topic is deriving observational bounds for effective higher derivative couplings in scattering events, as finding evi-

dence for higher derivative gravitational couplings would suggest a deviation from Einstein's theory [213]. It is also interesting to understand, for instance, to what extent it is possible to preserve double-copy structures in effective operators, see for example [214, 215].

Our pursuit of unknown physics from GWs has just begun, and we aspire to study gravitational attraction and its imaginable quantum elements with a strengthened theoretical effort utilizing effective QFT methods and novel observations. As discussed in Panel 7, *S*-matrix methods are not the only way to think about QG, but it stands clear that the EFT treatment of gravity provides a well-defined perturbative quantum theory of gravity, that allows a solid and universal background for facilitating potentially groundbreaking discoveries also in the quantum realm of gravity.

### 3.4 Mariana Carrillo González: *Scattering amplitude approaches for gravity*

The correct approach to defining QG remains an open question. One possibility, discussed in Panel 7, is to define it purely through an S-matrix approach, which involves understanding the fundamental and asymptotic states of the theory, whether they are point particles or strings and branes. This method has the advantage of utilizing well-defined mathematical techniques and can effectively capture the fundamental states, including resonances, of the theory. However, it also presents significant challenges. Notably, many spacetimes of interest, including our expanding universe, do not support the existence of an S-matrix. In such cases, one could argue that an alternative description, such as a Lagrangian, might be inferred from the S-matrix, and this could then be used to define the theory in an expanding universe. Yet, this approach has a major drawback: It suggests that the S-matrix is not the most fundamental description of the theory, as it does not permit a universal definition across all physical backgrounds.

From a phenomenological perspective, it is evident that certain situations do not require an S-matrix approach. As previously mentioned, in our expanding universe, an S-matrix cannot be defined, but this does not imply the absence of observables. Correlation functions on spacelike surfaces are well defined and have been both computed and measured for various observables, such as the temperature and polarization of photons from the CMB, as well as matter or galaxy overdensities. These measurements have provided a wealth of information about the content and evolution of our universe, all without requiring an S-matrix.

There are also scenarios where the S-matrix approach might unexpectedly prove beneficial. For example, in the dynamics of binary systems of compact objects during the inspiral phase [216]. While the S-matrix approach involves hyperbolic encounters, whose waveforms are more challenging to detect, it is known how to map these results to the bound case, provided that only local-in-time effects are considered. However, whether tail effects, which give non-local-in-time contributions due to gravitons that travel long distances before backreacting on the binary system, can be captured by this approach remains an open question. It would be intriguing to explore other astronomical processes in which S-matrix techniques could be applied effectively. Furthermore, it is worth investigating whether an S-matrix can be defined in dS space [217, 218], which would extend the applicability of positivity bounds to these curved spacetimes. On the other hand, alternatives such as causality bounds can also provide insight on the physical Wilson coefficients of EFTs on curved spacetimes [219–221]. These techniques analyze violations of causality at low energies on non-trivial backgrounds. An advantage of causality bounds over other methods, such as positivity bounds, recently reviewed in [93], is that they can be applied directly to cosmological backgrounds and higher-point interactions. This is a promising direction for understanding bounds on higher-derivative gravitational operators without many additional assumptions required by other approaches.

### 3.5 Bianca Dittrich:
### *Quantum spacetime and quantum gravity*

The discussions in Panel 1 revealed different aims and viewpoints of what a quantum theory of gravity should aim to achieve. Non-perturbative approaches such as CDT [222], CST [223] and spin foams [224] aim firstly to explain the emergence of a quantum spacetime (QST) from some form of fundamental constituents. For other EFT-inspired approaches, the focus is rather to construct a quantum dynamics of the gravitational field.

Starting with very fundamental ingredients, e.g. partially ordered sets, as in CST, the task to show that in some kind of many-element limit there emerges a notion of a smooth manifold is highly involved. One then also needs to provide a construction of geometric variables to extract the quantum dynamics of these variables.

By contrast, with a notion of a smooth manifold and the length metric as a fundamental variable already at hand, one has much more direct access to the dynamics of this length metric.

Spin foams seem to be able to bridge both viewpoints. Spin foams provide a Lorentzian path integral description based on a rigorous notion of quantum geometry. This notion of quantum geometry allows for different notions of (kinematical) vacua and fundamental excitations of quantum geometry [225–227]. A key open question is then whether one can construct (physical) quantum states which can be interpreted as approximate notions for smooth spacetimes [228]. This task is quite similar to investigating phase diagrams and condensation phenomena in condensed matter systems. This is very challenging for the four-dimensional spin foam models due to their highly involved amplitudes. But simplified analog models have shown rich phase diagrams and thus phase transitions, where we expect interesting continuum physics to emerge [229–231]. Recent work has also shown that a Lorentzian simplicial path integral approach such as spin foams [232] can avoid the conformal factor problem, which prevented the emergence of smooth spacetimes in many Euclidean approaches.

Assuming such (physical) quantum states peaked around "background fields" describing approximately smooth geometries do emerge, one can perturb around such configurations and extract the dynamics of such perturbations. To this end, it is very helpful that the theory does already include geometric observables. Indeed, it has been shown that such a perturbative continuum limit of spin foams does lead to (linearized) GR to leading order, and a Weyl-squared correction to sub-leading order [233,234]. This Weyl-squared correction can be traced back to a quantization-induced enlargement of the configuration space from length to area metrics [235, 236]. Spin foam dynamics in the continuum limit is therefore rather described by effective actions of area metrics rather than length metrics [237]. This illustrates how starting from a deeper level of quantum geometry, one arrives at a richer playground for the effective quantum dynamics.

### 3.6 Paolo Di Vecchia:
### *This is what I learned from the different approaches to quantum gravity*

The organisers of the program on QG at Nordita made a great effort to bring together people who work on different approaches for constructing a consistent quantum theory of gravity. I separate them in two groups. In the first group I put the approaches based on CDT, on CST, on tensorial GFTs and on spin foams. In the second group I put the approaches based on ASQG, on non-local gravity theories, on quadratic gravity and on swampland and ST. This program made me to realise that the two groups practically do not talk or even try to connect to each other. The reason, I guess, is that they use quite different techniques. In principle one could try to compare the physical observables that one gets in various approaches, but, as far as I understood, this is not easy to do at this stage. I was also surprised to hear that those in the

first group are not able to see gravitons to appear in their approaches, if I understood correctly. Although I am more familiar and more incline to work on the second group of approaches, I think that also the research of the first group should be pursued further in order to compare it with other approaches, with the semiclassical approach to gravity and eventually with the observations. At least that is what I understood from the seminars and the various discussions and panels.

It is more clear to me how instead the second group of approaches can be connected among themselves and eventually also with experiments. For instance, in the case of ASQG, one gets information about QG at the Planck scale and then, using the RG flow, translates this information at present energies. In practice, the aim of the second group of approaches is to arrive at an EFT Lagrangian that can be, in principle, compared with the one obtained by the various approaches and eventually with the experiments. I have not understood that this is also the aim of the other approaches, but this may be due to my ignorance. Anyway I do not want to give the impression that the most important thing for a theory to be considered is its connection with experimental data because we have various examples in the past of theories developed for trying to explain some phenomenological aspects, but then later became relevant for other things, as for instance Yang-Mills theory.

In this connection I was especially surprised about many strongly negative statements concerning ST. I agree that ST did not live up to many original statements made by many people working with it, but one cannot deny that it has had and continues to have a very strong and positive impact on various aspects of theoretical particle physics.

In conclusion, I enjoyed a lot to be part of this program for three weeks and now I have a better understanding of the different approaches used to get a quantum theory of gravity. It would be nice if, in the near future, the various approaches could converge providing results that could be compared with each other and, even better, if they could be compared with experiments.

### 3.7 John F. Donoghue:
#### *Quantum gravity as a QFT*

QG starts its life at ordinary energies as a QFT with the metric as the dynamical variable [125, 126,238]. There are frontiers at either extreme of the energy scale. At low energy, there could be a frontier of quantum theory in the area of macroscopic physics, leading to insights about decoherence and the emergence of classical physics. This would likely apply to all interactions, but gravity could be the leading indicator. At high energy, we expect all of our fundamental interactions to be modified by new physics, and here gravity is the most pressing interaction calling for a modification.

The workshop focused primarily on the potential high energy modifications to gravity. During the week that I was in attendance, there was a healthy representation of many of the proposed alternatives. As a benefit of the heavy dose of discussions at this workshop, it was interesting to see everyone being thoughtful about their motivations and aspirations for their own chosen direction, and also candid about their perceptions about limitations of their own and other approaches. It is everyone's hope that by considering the various theories we could potentially learn about common features which might be universal in QG, even if we cannot experimentally prove any of them to be correct. From the discussions at the workshop, I am less convinced of a universal convergence, but there may be different branches of the theoretical landscape. Moreover, there could be yet other possibilities which are not actively explored at present, such as the idea of Random Dynamics [239] which I commented on during Panel 2.

My own contribution was on theories in which the metric remains the primary dynamical variable even in the fundamental theory, in particular quadratic gravity [3, 5] and ASQG [8, 240]. These provide somewhat different visions of renormalizable theories for QG, but provide

a solution to the proliferation of parameters for operators of higher dimension which is found in the EFT treatment. However, there is still the issue of guaranteeing that the high energy behavior of physical amplitudes is well behaved, and I highlighted this as an important frontier for the future of these studies. Despite the ability to limit the parameters of the gravitational theory, it is not clear what constitutes a successful conclusion for these theories.

It is also possible that this class of theories, with higher derivative kinetic energies, could lead to emergent causality, with violation at high energy but normal causality at low energy [49, 50]. As a phenomenologist at heart, it is my hope that this could lead to the elusive experimental signal for QG. In the discussions, several of us made the case that we could already have such a theory under consideration. Starobinsky inflation [52] is one of the few remaining variants which remains experimentally viable given the present bounds on the ratio of tensor to scalar perturbations, and which appears as plausible from the viewpoint of QFT. Moreover, for consistency the theory would have a second invariant besides $R^2$, which can be taken to be the square of the Weyl tensor, because these invariants mix under the RG flow [102]. This extra interaction generically leads to emergent causality. A key goal for the future would be to identify a signal for causality violation in the early universe.

### 3.8 Fay Dowker:
#### _Forks in the road, on the way to quantum gravity: Setting out our multiple paths at Nordita_

Meetings like the Nordita Scientific Program "Quantum Gravity: From gravitational effective field theories to ultraviolet complete approaches" are crucial in advancing our collective efforts as a community to achieve a theory of QG. I found it very helpful to put the perspective from CST to colleagues and to understand better where and why we disagree.

If one frames discussion of QG as a set of "forks in the road" [241], several such forks were represented at the meeting in week 2. For example in the first talk, John Donoghue set out a useful framing of one of the forks. He described the non-renormalizability of perturbative QG and then mentioned the analogy with Fermi's four-fermion interaction as a successful effective theory which is non-renormalizable and which breaks down at high energies. Faced with this singularity at a particular scale, one can either try to improve the predictability of the theory and extend it to higher energies beyond the scale of the breakdown by modifying the interactions keeping the degrees of freedom the same, or one can take the breakdown of the theory as an indication that truly novel physics becomes relevant at the scale at which the old theory breaks down. In QG we do not yet know what branch of Donoghue's fork in the road we will have to take, and both directions were represented at the meeting. CST takes the "new physics" direction at that fork and proposes "discrete causal relations" as the new degrees of freedom.

Another fork in the road involves matter. Full unity can only be achieved in a theory of spacetime and the SM together. But today, there is a fork that is a decision about whether we can make sense of a quantum theory of spacetime alone and wait till later to include matter or whether the two must necessarily go together and both views were expressed and argued for: ST is the epitome of an approach that aims for a theory of spacetime and the SM together. In CST, in its current stage, we are trying to understand how to build a quantum theory of causal sets alone without additional degrees of freedom.

One issue we discussed is, I believe, common to all approaches is the cosmological constant problem which manifests itself in slightly different forms in different approaches but which haunts all our efforts. In CST it takes the form of what Renate Loll calls the "entropic threat" which is that the non-manifold-like histories dominate the path sum in the counting measure. Work by Steve Carlip and Sumati Surya and others has provided evidence that at least some of this threat can be overcome by a judicious choice of causal set action. Topology change is

another fork we discussed: There are approaches like CST that cannot help but incorporate it, approaches in which there is a choice, and approaches such as canonical quantum gravity in which one cannot include it. If it *is* included kinematically, one may find that it is dynamically suppressed and that was pointed out in the discussion. Unitarity is a fork that is very pertinent: Some approaches, e.g. CST, deny unitarity, some demand it and there is no consensus — that's healthy. Locality is a fork that was harder to discuss given it has several different meanings in context. CST is non-local but the discussion perhaps failed to pin "locality" down precisely enough across approaches for a fruitful discussion to take place.

The meeting brought clarity to areas of consensus and to areas of lack of consensus alike, both of which are useful. My view is that the latter is actually particularly useful. We should figure out where we disagree and sharpen our understanding of the reasons we take different approaches because at our current stage of development we as yet lack a rich store of empirical observational data to guide us and we would be wise to encourage a diversity of well-motivated approaches. We need to disagree and to keep talking to each other.

### 3.9 Gia Dvali:
### *Visions about UV completion of gravity*

For me, as a particle physicist, QG is defined as a quantum theory of a massless particle of spin two, which in the deep IR flows to GR. In my view, for a theory formulated in this way, the main challenge is to understand its UV completion, and in particular the role of BHs in it.

There is a strong evidence that in such a theory, the high-energy collisions at center-of-mass energies much larger than the Planck mass are unitarized via the formation of BHs. This is true both in GR viewed as an EFT of the graviton, as well as in its string-theoretic embedding.

By the very nature of BHs, this feature provides a certain UV-IR connection: The higher the center-of-mass energy is, the larger are the BHs. In this way, a deep-UV gravity is also a deep-IR gravity.

In my opinion, this feature provides one of the strongest indications that the consistent formulation of QG must involve the notion of asymptotic S-matrix states. This is a powerful selection tool, as it excludes any would-be classical background not permitting an S-matrix as a valid vacuum of QG. An important example of an excluded vacuum is dS.

At the same time, the above indicates that a formulation of QG that avoids this UV-IR connection must get away with BHs. If so, such a theory will not be able to reproduce GR in the deep IR. It therefore would not satisfy the definition of a quantum theory of gravity given above.

An important task for future studies would be to make the above concepts more rigorous.

### 3.10 Steven B. Giddings:
### *Quantum gravity*

The problem of QG has been with us for a long time, with different communities long working on quantization programs with distinct starting points. We had important summaries of latest developments in some of these at the workshop. But, in my view, it is important to take a step back and reconsider the nature of the problem and possible shape of the solution. As our knowledge of the problem has advanced, that has continued to give us clues. In the allotted space, I'll try to make some basic points regarding these, connecting to the very stimulating discussions at the workshop; for additional perspective, see [242].

A first question for QG is what is a good starting point; e.g. various difficulties seems to reveal a conflict between quantum mechanics and spacetime. In my view a reasonable starting point is with the end: We are plausibly looking for a *quantum mechanical* theory. One reason to emphasize this is that, while people might consider abandoning different principles of quantum

mechanics, it has so far been very difficult to do so in a realistic theory. Examples include generalizations to non-unitary evolution [243] or non-linear formulations [244], which in the end seem to lead to disastrous behavior [73,245]. The difficulty of modifying the principles of quantum mechanics, the so-far lack of a good alternative framework, and the fact that it has been experimentally tested in many domains, provide strong motivation for describing physics within a quantum-mechanical framework — whose tightness of structure may aid our quest.

In contrast, there seem to be ample reasons to abandon a fundamental description in terms of spacetime manifolds; one that is compelling is that it is hard to make sense of any manifold structure at distances shorter than the Planck scale, either from the point of view of theory, or of possible measurement. And, this lesson may well carry to longer scales.

These are some of the reasons I have advocated taking a "quantum-first" approach to the subject [30–32, 246, 247], as I briefly described in Panel 3. By this I mean, beginning with a suitably general quantum-mechanical framework (see e.g. [30]), and describing additional structure, e.g. of the space of states and observables, to well-approximate gravity in familiar regimes. If this is viewed as too abstract, consider that QFT — which apparently describes the rest of physics — can likewise be considered to be derived by starting with the principles of quantum mechanics, then implementing the principle of locality and special relativistic principles, primarily Poincaré invariance.

Even in the *approximation* where we perturbatively quantize GR, we have found important departures from the principles underlying QFT, such as a departure [43, 246] from its formulation of locality in terms of commuting subalgebras of observables at spacelike separations [248]. This begins to illustrate that the observables of QG are expected to have a very different algebraic structure, which one expects to connect to a different structure on the Hilbert space, corresponding to a significant modification of the locality principle. Developments continue in these very important directions [44,249–254], with the recent story of type II von Neumann algebras [255–257] being one interesting tip of the iceberg [258].

Finally, if we are to preserve quantum principles, a key one is that of unitarity, e.g. in scattering of states that asymptotically look like small perturbations of Minkowski space; I discussed this further in Panel 2. Such asymptotic states, with sufficiently high energies, can produce quantum BHs, whose evaporation seems to lead to a "unitarity crisis". If one is to save unitarity, it appears to require departures from conventional locality [259], which in view of the preceding may not seem as surprising. The question is to describe such departures; some initial description of an effective parameterization of corresponding interactions appears in [260,261], and references therein. Interestingly, the apparent necessity of modifying near-horizon physics also suggests the possibility of observational signatures [262–264].

Such effective parameterizations leave the question of the more basic quantum formulation of the theory — but it may be that we will not find that by quantizing classical structures.

### 3.11  Alessandra Gnecchi:
### *IR quantum gravity from the UV*

The success of the SM has motivated theorists to quantize gravity as a theory mediated by a fundamental spin two field. Unfortunately, despite the wonderful mathematical consistency of Yang Mills theories, GR as the theory of a spin two field simply cannot be renormalized. Throughout the workshop, talks and discussions have highlighted how QG is a very concrete framework of research that continues to challenge our description of the universe and is bringing important insights towards extensions of the paradigms of GR. What is then, the GR completion to a theory of QG in the UV? Several proposals exist, that modify gravity in some local and some non-local ways, in some continuous and some discrete ways. Despite the ultimate quest for a unique UV-complete gravitational theory, an EFT approach to QG is needed to better identify the fundamental properties of gravitational interactions.

Among the approaches currently studied, the swampland program is the systematic investigation of the consistency of an EFT once we couple it to gravity. Recently associated to the swampland program within ST, this approach refers in general to the attempts at identifying the breakdown of an EFT from constraints coming from gravitational interactions. In this context, ST offers a unique point of view: Being a UV complete theory of gravity, it allows us to build the EFT from a microscopic construction, and track the breakdown of its validity to a more fundamental principle in the full theory. Since gravity has a unique feature of coupling the UV and IR physics, generally referred to as UV/IR mixing, ST can impact the EFT in some specific way. The question of defining which EFT is actually part of the *landscape* or instead part of the *swampland* acquires thus a well defined meaning in ST, and has been formulated in terms of *swampland conjectures* over the recent years [67, 68], including a *BH entropy distance conjecture* [265, 266]. An important question now arises: Is the ST landscape unique? Are other gravitational extension of GR yielding a different swampland scenario? Are there other gravitational EFTs that can be consistently extended to the UV but are not part of the string landscape? The authors of [83, 86] have started addressing this question, which was also addressed during the panel Panel 5.

No matter the extension of classical GR one may choose to explore, it has to eventually include a resolution of the BH singularity and the unitary description of semiclassical BHs, as discussed in Panel 12. From an EFT point of view, BHs are the gravitational states appearing in the spectrum of the theory as soon as it is coupled to gravity. They are at the origin of the first bounds on quantum fields like the Bekenstein bound [267] and the UV/IR mixing CKN bound [268]. Despite evolving in a semiclassical regime, evaporating BHs cannot be explained in a satisfying way without understanding properties of gravity and the gravitational path integral beyond the Gibbons-Hawking approximation. In the framework of ST, the AdS/CFT correspondence has offered a novel way to formulate the problem of information in the context of BH evaporation. The evaporation of BHs at late times is characterized by a novel entanglement structure, that has been quantified holographically by extremal surfaces call *island surfaces* [154]. Their gravitational counterpart appears to be a new saddle point of the gravitational action, dominating after the Page time, corresponding to wormhole geometries, which sparkled interest recently in the investigation of Euclidean saddle points of the gravitational action. Wormholes are becoming more and more relevant in understanding the evolution of our universe [269].

The efforts made by the organizers, in bringing together experts from diverse angles of research in QG, has created a lively atmosphere during the workshop where participants could confront different approaches and learn each other's progresses. Such a successful scientific interaction needs to be supported also in the future and reach more people in the QG community.

### 3.12 Renata Kallosh:
### *Future research directions in the field of my interest*

The future of the whole field of QG at this point in time is particularly difficult to grasp since we do not expect important high-energy particle physics experiments any time soon. New ideas about the symmetries beyond those in the SM, which would be supported by the experiments, maybe would give us some hints about high-energy physics, including gravity. But these are out of reach.

On the positive side, many new experiments/observations are expected in cosmology in the next couple of decades. Therefore the issues of what we believe is a fundamental theoretical physics, including gravity, will be tested in the relatively near future.

There is a general understanding that (super)-ST is the best theory available which includes QG in a consistent way. The problem of deriving the consequences from (super)-ST

and predictions towards observational cosmology stem from the fact that ST is defined on 2D world-sheet, but the observations are performed in 4D space-time. To bridge these two is a serious problem, namely, one has to come up with something like ST inspired 4D supergravity first. Nevertheless, one can still try to use the best part from ST ideas and its symmetries towards cosmology. For example, in our current research [270] we study the concept of the "target space modular invariance", which can be applied to observable cosmology, as opposite to duality symmetry in ST which acts in 2D and is not useful for properties of spacetime directly.

The swampland ideas have their negatives. However, from my perspective, they have added to our current concerns about the future of theoretical physics something positive. It is a simple proposal to make various ideas from ST compatible with already available observations in cosmology. The recent proposal by Casas and Ibanez [81] suggests to study modular invariant cosmological models with potentials which have a plateau at large values of the inflaton field. We are developing these ideas and the important point here is that these models are falsifiable by the future cosmological observations.

Of course, it is important to understand to which extent we can rely on CMB physics data without taking into account the $H_0$ tension problem. Here we have to wait for more data.

Assuming that the data from Planck satellite will survive, the future LiteBIRD satellite, to be launched in 2033, will tell us about the primordial GWs from inflation. Will they be detected at the level predicted by $\alpha$-attractor models [271] and their generalizations involving target space modular invariance? There is, of course, a long waiting time here, nevertheless there is a perspective that there is a set of theoretical ideas about QG which will be probed by the experiment. Many countries now joined the LiteBIRD mission, for example France had a LiteBIRD day meeting in May 2024. And of course LISA mission, to be launched in 2035, will offer huge opportunity to test QG ideas, in cosmology and beyond.

Many theoretical ideas about QG were discussed at the workshop, and of course, all of them should be supported in their pursuit of establishing internal and mathematical consistency. But it is useful to keep in mind that eventually some ideas about QG will be tested in future experiments.

### 3.13 Alexey Koshelev:
#### *Path to quantum gravity*

Following a brilliant occasion to give a talk, to participate in panels, and to speak to many experts and friends during the Quantum Gravity 2024 program at Nordita, a natural question to discuss is: What is QG?

I would advocate the viewpoint that a QFT approach to QG seems solid. It may not be the final point in our construction. ST, emergent spacetimes, and other ideas may take over, but for the time being we know that QFT works for all but gravity, and gravity classically is no worse and not exceptional compared to any other theory. It will be then quite natural to assume QFT as the guideline, or at least to give a serious try to construct QG using standard methods and techniques of QFT. I really enjoyed following different arguments in Panel 1 on exactly that matter, still staying on the point that QFT of a massless spin-two field should be at least an intermediate step in our way to a complete unveiling of the gravity nature.

It is however understood that a consistent QFT for gravity requires infinite towers of derivatives [272] unless you are ready to break something like Lorentz invariance or diffeomorphism invariance, etc. This approach is heavily motivated by string field theory where nonpolynomial derivative operators arise naturally, and manifest that strings are extended objects. Infinite derivatives are simply algebraically the only way to save a gravity theory beyond GR from ghosts like the one in quadratic gravity. The renormalizability of such theories is still a subject of intensive study, but surprisingly again infinite derivative operators are absolutely

needed to accomplish a construction of an asymptotically safe gravity. This provides an extra convincing reason to look closely for infinite derivative theories.

This approach is not benign and has certain unexpected features. Infinite derivatives lead to new and strange degrees of freedom around non-flat backgrounds, prompting the question of unitarity. This problem can be elegantly resolved in a string field theory motivated approach [176, 273]. Also in general one may expect that causality is at danger here. This however does not seem to be a problem, because we expect that even if such a violation will take place, it will be unresolvable [219], that is the one happening at scales hidden by the uncertainty principle. The reason for this are simply the scales in the game given that any QG effects are naturally anticipated at nearly Planck energies. One can go further and study causality and unitarity bounds considering scattering amplitudes. Such studies are on the way and got boosted thanks to new inspirations from the Nordita program, but are definitely very cumbersome and require time. Computing the Shapiro time advance (or delay) in infinite derivative models is a question sparked and discussed during the workshop with several experts, and this can be yet another test for these models.

The above problems were discussed in relation to different approaches to QG in Panel 2 and Panel 3, with an even more stringy bias in Panel 4. And I got a pleasant feeling that studying infinite derivative, i.e. non-local gravity theories, further may open new connections between QFT, strings, and many other paradigms.

However some questions which one would expect as answered or at least understood, are still not. For example: Why in majority do we accept a metric signature as given and often avoid considering a path integral with a totally arbitrary metric, releasing the metric signature constraint? A more tricky but still a viable question is why a spacetime dimensionality (if we assume to have a spacetime already though) is usually a fixed positive integer? Another major question which comes in is: If gravity gets a non-local formulation, should matter stay local or not? These puzzles may form a long and comprehensive study in the near future.

At the end I would like to thank the organizers for their immense efforts in bringing together experts from different schools of thoughts to discuss, to argue, and to seek for connections in a very diverse landscape of approaches to an outstandingly complicate problem of constructing a QG theory.

### 3.14  Renate Loll:
*The importance of being non-perturbative (and Lorentzian)*

The great diversity of viewpoints articulated by the workshop participants underlines the fact that QG means many different things to different practitioners, a status quo that reflects the scarcity of objective "reality checks" of what may constitute a valid formulation of QG, not to speak about the absence of testable new predictions. In a recent presentation at Radboud University, Armas reported on his effort of sifting through the "Conversations on Quantum Gravity" [274] to compile a joint wish list of all the problems a theory of QG should be able to solve, according to the volume's 37 contributors. The resulting list extends over several densely filled pages, indicating a serious overload of unrealistic expectations of what QG will be able to deliver — if only we knew what it was.

An uncommon but instructive way to frame a smaller and more realistic set of expectations is to examine the power and limitations of our most advanced theoretical and computational tools, and to what extent they have enabled us to understand the non-gravitational interactions, aka the SM, an arguably much simpler theory. This provides concrete benchmarks for what we can hope to achieve in terms of "solving" QG in the foreseeable future, without relying on the discovery of a hitherto unknown symmetry principle, or the emergence of a "next Einstein" with an advanced intuition of physics at the Planck scale. We must look specifically at methodology that has led to a quantitative understanding of local dynamics in a fully non-

perturbative setting, the most glaring missing element in our search for a fundamental theory of QG. A natural blueprint is provided by the strong interactions, where sophisticated lattice QCD methods allow us to perform first-principles computations of the theory's spectrum, in what constitutes a major, ongoing success story of theoretical high-energy physics [275].

Implementing a similar strategy for QG has proved a formidable challenge, since the dynamical nature of quantum spacetime turns out to be incompatible with the use of standard, fixed lattices, and also complicates the Wick rotation, essential for the application of powerful Monte Carlo methods. Fortunately for us, longstanding efforts by the community to make sense of lattice QG have finally produced a working version that solves these problems, where CDT [9,222,276] play a role analogous to that of holonomy variables in Wilson's breakthrough formulation of lattice QCD [277]. CDT lattice QG has been thoroughly tested in state-of-the-art Monte Carlo simulations, yielding spectral properties of several diffeomorphism-invariant observables in a Planckian regime, which indicate the emergence of a dS-like behavior from pure "quantum foam" [278,279], unprecedented in any non-perturbative formulation. These encouraging developments — against the backdrop of a complex landscape of QG candidate theories — point to some key features which in my view will be hallmarks of successful QG research in the near future:

(i) **Unique source of novelty for QG.** New quantitative information about fundamental QG and quantum spacetime will come from numerical "experiments" using lattice simulations. As non-perturbative, first-principles tools with a minimal number of free parameters, they can serve as primary "reality checks" of both Planck-scale physics and the presence of a classical limit. New insights are unlikely to come from dimensionally or symmetry-reduced toy models, which lack decisive aspects of the full theory [280]. The same holds for exact mathematical methods like algebraic QFT, say, which cannot describe the renormalizable QFTs of the SM either.

(ii) **Unique power of numerical methods.** Just like in the strong-gravity sector of classical GR, dedicated numerical methods are needed to extract the physical phenomena implied by the theory's dynamical principle, in GR given by the Einstein equations and in QG by the non-perturbative path integral. Since we cannot solve strongly interacting four-dimensional QFTs analytically, numerics are essential and QG formulations amenable to efficient, large-scale simulations/computations have a huge advantage.

(iii) **Primacy of QFT.** Uniqueness and therefore predictivity of QG is the elephant in the room of ingredient-rich, beyond-QFT approaches like ST but also of models with fundamentally discrete "building blocks" like CST. By contrast, non-perturbative QFT offers RG mechanisms to obtain universal continuum physics from a (lattice-)regularized formulation [281]. This has already produced nontrivial results, like the anomalous behavior of the spectral dimension of quantum spacetime first discovered in CDT [282] and later found in ASQG [283], without the need for exotic ingredients.

(iv) **Disappearance of "approaches".** In the vision just outlined, there will be an essentially unique theory of QG, which is the minimal non-perturbative quantum field-theoretic extension of GR. The community's focus will shift from arguing about "approaches" to developing diverse technical tools to optimally address the physics of QG in terms of suitable observables and their relation to phenomenology, where early-universe physics may be our best bet. Solving QG's conceptual challenges and calibrating our expectations of what the theory can deliver will benefit greatly from working with concrete computational implementations, amounting to an altogether optimistic and exciting outlook!

### 3.15 Leonardo Modesto:
### *The quantum gravity war*

The debate about *what QG is* has been going on for more than half a century to the point that the initial problem seems to have been forgotten. Moreover, some motivations for QG are actually incorrect. For example, QG is supposed to provide a mechanism for avoiding the spacetime singularities present in almost all solutions of Einstein's gravity. However, the singularity issue is not present at the classical level in Einstein's gravity because the latter is secretly Weyl's conformal invariant. Actually, the singularities show up at quantum level because of the conformal anomaly. Thus, contrary to what is commonly believed and written in most of the scientific literature, the quantization of Einstein's theory introduces a problem that is not present at the classical level.

Therefore, we consider to be useful to recap what the problem is and how it can be addressed in the most conservative way possible. First of all, gravity can be quantized exactly like all the other fundamental interactions in the framework of *QFT*. In this procedure, there is no inconsistency just as there is nothing wrong in the quantization of Fermi's theory of weak interactions. Of course, similarly to Fermi's theory, Einstein's gravity is perturbatively a *non-renormalizable* theory. Therefore, we can define the path integral for gravity consistently with gauge invariance, and, from the partition function, we can get the quantum effective action and all the *n*-point Green's functions. Finally, the Hilbert space of the theory can be built through out the *reconstruction theorem*. To conclude, the construction of the above *is QG*. However, when we try to compute an amplitude beyond the tree-level approximation in the loop-perturbative expansion, Einstein's gravity turns out to be a non-renormalizable theory, namely we cannot hide the *infinities* in a redefinition of the classical action. In the case of weak interactions, the very same issue was addressed replacing at first the Fermi's action with the so called intermediate vector boson model, and the gauge theories with spontaneous symmetry breaking afterwards. In gravity, we have to follow the same path and propose a new gravitational action consistent with the *renormalizability guiding principle* and unitarity. To our great surprise, in the last fourteen years, local and non-local theories beyond the Einstein-Hilbert action principle have been (re-)discovered and extensively studied. Such theories are defined by an action, and/or equations of motion, provide all the same solutions and stability properties of the solutions of Einstein's gravity, are super-renormalizable or finite at quantum level, and unitary at any perturbative order. Indeed, they satisfy the same Cutkosky rules of Einstein's gravity coupled to the SM. In cosmology, such theories provide a simple, falsifiable, and universal non-inflationary scenario that sees the big bang replaced by a Weyl-conformal phase of gravity.

Hence, we do not need to quantize extended objects, or other non-perturbative and/or emergent approaches to make gravity enjoy the very same or better properties of the other interactions. Actually, we can claim to have a finite theory of all fundamental interactions in the QFT framework. Of course, other proposals are very welcome and sources of stimulation for creativity, though we must be aware that they are not so needed.

What we have to investigate in the future is why we have so many consistent theories (non-local and local) of QG. As a way out of this puzzling issue, we are now proposing a change of paradigm. In QG, as well as in condensed matter physics, we should focus more on the universality classes rather than the details of the action or the equations of motion. In order to sketch such idea, let us assume a fundamental theory to be free of Landau poles, and to be super-renormalizable or finite. Hence, if the theory is non-local, we will very likely observe scattering amplitudes soft and smooth in the UV regime. On the other hand, if the theory is local, other peaks corresponding to purely virtual ghosts will show up in the amplitudes observed at a hypothetical collider. The third possibility is the only renormalizable and unitary theory of gravity, namely quadratic gravity.

### 3.16 Daniele Oriti:
#### *Sparring perspectives*

I want to articulate a general impression I gathered from the workshop. Thirty years ago [284, 285], a clash of two perspectives was visible, manifested in a rather harsh clash of communities; thirty years onwards, thanks to dedicated efforts [286], we see an evolved version of those clashing perspectives, now more sympathetically sparring, and less attrition between communities.

One perspective can be traced back to the GR/gravitational physics tradition, the other to the QFT/particle physics tradition, but the difference is not anymore in tools or expertises, which are mostly shared across the board.

The older versions of these perspectives were: One, speaking the language of EFT, for which gravity is only gravitons and the gravitational force (I am exaggerating), thus closer to background-dependent, perturbative physics, and only taking seriously QG results formulated in this language; the other aiming directly at a non-perturbative formulation or at the identification of fundamental structures "constituting" spacetime [287], thus having more difficulties extracting perturbative physics and making contact with phenomenology, and only taking seriously QG formalisms that do not use a perturbative language and that tackle head-on foundational questions.

The two evolved perspectives have not so much to do with the dichotomy "perturbative vs non-perturbative": Everybody agrees that QG should ultimately be formulated non-perturbatively, and that we should make solid contact with phenomenology (now, we know we can [145]). They differ about which principles we deem necessary in the future foundations of QG, and which ones we should (reluctantly) drop.

One perspective holds that the main lessons of GR (dynamical spacetime structures, diffeomorphism invariance, generality of allowed boundary conditions), carried to the quantum domain, force us to conclude that unitarity can only be approximate and relational (referred to specific physical clocks); that causality, like geometry, is dynamical and subject to quantum fluctuations (when not emergent [288]); that locality with respect to the manifold is meaningless, because of diffeomorphism invariance, and approximate only when defined with respect to physical frames, because they are also quantum and dynamical; thus, those principles should be ultimately dropped, and we better start immediately developing QG formalisms that do not rely on them.

The other perspective holds that those principles, at the root of the success of the SM and of most applications of GR itself (based on special solutions or boundary conditions), are the only conceptual basis we have for advancing; therefore, since the clash with GR is there, we have to renounce those lessons as central to QG; since we cannot be confined to special GR solutions and perturbations around them (e.g. to solve BH puzzles), only special boundary conditions (e.g. asymptotically flat or AdS) should be considered, those that allow (at least on the boundary), a precise notion of locality, causality, unitarity; this leads to taking AdS/CFT as a complete definition (not just a sector) of QG, or to expecting full QG to be a collection of such "holographic" correspondences for different (suitable) boundary conditions.

I do not claim any equidistance. One perspective is correct, I think, and more fruitful in the long-run; the other may be productive in the shorter run and a step to realize the other. But this sparring of perspectives is not a problem, if it is about priorities or heuristic strategies; neither is it a problem if it is about how QG will ultimately look like (we will pursue each direction, and let the combination of theory and observations judge the result). It is a problem, however, if it turns into an ideological barrier, preventing attention to respective merits, or communication across it. For example, it could induce a dangerous neglect for the insights of the AdS/CFT correspondence, or a limited attention to the quantum consequences of tying gravity to other interactions, on one side, and, on the other, a dangerous neglect for

strategies to solve foundational puzzles arising when dropping key pillars of standard QFT, or for their implications for quantum foundations.

We need a single, plural community, within which any such clash can be resolved or at least be put to maximal fruition. As also this workshop shows, we now see this community forming (it is an ongoing process: ISQG).

### 3.17 Jan M. Pawlowski:
### *Asymptotically safe gravity in a nutshell*

Amongst the many contenders for a consistent theory of QG, ASQG has the appeal of its conceptual simplicity, defining QG as a QFT. In its simplest form, it is the QFT of a dynamical metric field paired with diffeomorphism invariance. Then, the requirement of a UV completion of gravity as a QFT entails the existence of a stable UV fixed point. The existence of such a fixed point is the defining property of any UV-consistent QFT, an important example being non-Abelian gauge theories with asymptotic freedom, a UV-stable Gaussian fixed point. By contrast, QG requires a non-Gaussian interacting fixed point, the Reuter fixed point [289]. Proving the physical viability of this setup leaves us with several essential tasks, that have only been completed partially by now: **(i) Existence:** Does QG admit a stable UV fixed point? **(ii) Unitarity:** Does ASQG satisfy unitarity bounds? **(iii) Asymptotically safe particle physics:** What is the set of asymptotically safe particle physics models (SM and beyond), and what are their predictions? **(iv) How to:** What is the status concerning the systematic error control in asymptotically safe computations? Below we will briefly summarise the state-of-the art, and for more information we refer to [290, 291] and in particular to the ASQG part [133, 292–297] in the "Handbook of Quantum Gravity" [2] and references therein.

(i) **Existence.** Since the seminal work of Martin Reuter in 1996 [289], an overwhelming amount of non-trivial evidence has been collected in hundreds of works, that in combination solidifies the existence of asymptotically safe pure gravity as a QFT, see e.g. [2, 290, 291] for a compilation of different works and their assessment. Roughly speaking, the key question concerns operators whose insertion into the theory may destroy the UV fixed point: In short, all orders of $R$ and $R_{\mu\nu}^2$ have been considered as well as the full momentum dependences and combinations thereof. All these operators exhibit a close-to-perturbative scaling, except the Newton constant that necessarily has a large anomalous dimension. While certainly not a proof of asymptotic safety, this pattern provides overwhelming evidence for the existence of the Reuter fixed point.

(ii) **Unitarity.** Given the mature state of the existence question (i), this is the decisive one left. Most ASQG computations to date are performed with a Euclidean spacetime signature, but first promising direct results with Lorentzian signature exist. In particular, the results include indications of the the absence of Ostrogradsky ghosts in the graviton spectral function. These computations are accompanied by results on scattering amplitudes in line with unitarity bounds, see [294, 296]. Fully consolidating these promising results would present a major step towards establishing ASQG as a unitary theory of QG.

(iii) **Asymptotically safe particle physics and EFTs.** The existence, stability and physical signatures of gravity-matter systems have been studied for more than two decades, leading to a plethora of results including low energy constraints and predictions, see [292, 294, 296]. Full reliability requires the solution of large systems of coupled partial differential equations for correlation functions. In short, a comprehensive survey of UV-complete models has been launched and already shows very promising first results.

(iv) **How to:** The resolution of the tasks (i)-(iii) requires a computational approach with reliable systematic error estimates. This is a chiefly important but unfortunately highly technical question, as the assessment of the systematic error analysis in any numerical approach requires a minimal technical familiarity with the approach. By now, the functional RG approach, that is predominantly used for computations in ASQG, can draw from four decades of advances and benchmark results in strongly correlated non-perturbative systems, including some that bear a close resemblance to ASQG. We refer the reader to [290, 296] and [298]. In the latter work, the concept of apparent convergence and the modular structure of functional RG equations (LEGO® principle) is evaluated.

### 3.18 Roberto Percacci:
####     *Layers of quantum gravity*

When one says "Quantum Gravity", one presumably means a quantum theory explaining why and how apples fall. Such a theory already exists: It is called the EFT of gravity. It agrees with GR in some regimes and predicts small deviations from it in the limits where GR is no longer expected to be accurate. There is little reason to doubt its predictions.

Still, for different reasons, many physicists are not completely satisfied with the EFT. On one hand, the EFT has its own limitations and there are interesting questions that it does not answer. On the other, its foundations are utterly different from those of GR. In fact, it can be thought of as a quantum version of the older theory of Newton, because in the EFT, gravity is treated as a force and not as the geometry of spacetime. This makes it all the more remarkable that it can also predicts the classical effects of GR.

Research falling under the QG umbrella is split in a large number of sub-fields, but there is one main subdivision that emerged clearly during the workshop: There are those whose main aim is to compute scattering amplitudes (a modern version of falling apples) and those who insist on looking for a quantum version of GR. I suggest that QST would be a more appropriate term for what the latter group is trying to achieve.

For someone with a QFT background, it may not be so evident that a theory of QST is needed at all. I think that a good motivation comes from metric-affine gravity (MAG) theories, that extend GR in the classical domain, making it more similar to the theories of the other interactions. In this context, gravity as we know it is seen to be in a Higgsed phase. The metric plays the role of the Higgs field (the order parameter), so the QST question "what is the origin of the classical metric?" is seen to be an exact gravitational analog of the question "what is the origin of electroweak symmetry breaking?". In the case of gravity, one may well have to go beyond QFT to answer it.

The discussions have also highlighted that we can view theories of QST as the "lower layer" of QG, and the study of scattering as its "upper layer". Understanding the former will require non-perturbative methods. As in the theory of the strong interactions, once a condensate is shown (or assumed) to exist, one can study its dynamics by perturbative methods. This is the place for the EFT of gravity, and for possible QFT-based UV completions. Remaining within perturbation theory, one such approach is based on Lagrangians quadratic in curvature. Some progress in this direction has been reported, but it emerged that a better understanding of scattering states in these theories is needed. On the other hand it is possible that also the upper layer will have non-perturbative aspects, as for example in ASQG. Whether and how these are related to the lower layer is an interesting question. In its continuous formulation based on the functional RG, ASQG seems ill equipped to explain the emergence of spacetime. On the other hand, CDT can be viewed as a way to prove ASQG by discrete methods, and is very clearly a theory of QST.

Of the other approaches to QG, QFT and ST deal with the upper layer, LQG and CST with the lower one. Because their main goals are so different, it makes little sense to compare their achievements. A classical spacetime is a major goal for the lower layer and merely a starting point for the upper layer. In principle one may expect that the lower level, more fundamental theory should be able to explain also the phenomena of the upper layer, but that may be too much to ask for. In strong interaction physics we have come to accept that high energy and low energy phenomena are described by different theories. It is quite possible that something similar will happen for gravity.

### 3.19 Lesław Rachwał:
### *Quantum gravity is great!*

QG is, without any doubt, a very fascinating topic of research in theoretical physics. By some was even heralded as the Holy Grail of all fundamental physics, but surely it is one of the most complicated and difficult part of our theoretical description of nature. Moreover, despite the 70 years of the continuous and arduous efforts of the best minds among physicists and mathematicians, still we have not reached a consensus on how this theory looks like, which mathematical apparatus it uses and in which theoretical physics framework could be the best expressed. Also on the bad side, we have not seen any direct quantum gravitational experiment confirming any of the models or the approaches studied vastly in the literature. Probably, there is not much hope for this to happen in our lifetime since the Planck scale $E_{\mathrm{Pl}} \sim 10^{19}$ GeV is much bigger than the other typical scales in the SM.

However, (almost) everyone of us is quite sure that gravitation has to be quantized (in one way or another). We only differ in using different (sometimes complementary, sometimes contradictory) approaches to QG. It is not true, contrary to what is conveyed often to the public, that there is a fundamental clash between quantum mechanics and gravitational theory. This junction works pretty well in low-energy effective quantum gravitational theory [299]. Of course, the most interesting and most difficult problems arise in the deep IR and UV regimes of the QG domain of description of the quantum universe. In the literature one can find many different methods of quantization, heavily proliferated number of models and theories of QG within various approaches. We often propose to modify the rules of gravitational theory, while keeping quantum mechanics intact, to avoid "gravitization" of quantum mechanics.

In the last few years we have experienced a spectacular success and developments on the theoretical side of QG with the creation of many models which are claimed to be successful in solving all the previous purely theoretical problems like unitarity, renormalizability, UV-finiteness of QG and singularity problems of GR. It is a pity that, for example, for the problem of unitarity of local higher-derivative gravitational theories there exist many different solutions (proposed by Anselmi-Piva, Mannheim, Modesto, Donoghue-Menezes, Salvio, Buoninfante, Shapiro, Asorey [3, 167, 300–305] and others), but we do not see a convergence, and some theoretical attempts at unification or mutual exclusion of them are strongly needed here as based on the reasoning in Panel 2 (with the notable exception of convergence of the fakeon and the multi-norm approach of Salvio that so far have been shown that some of their testable predictions actually agree [306]). We would like to see in future these apparent differences to disappear or to evolve into the most clear experimental signatures to test them.

In QG we still face a lot of conceptual questions that were also discussed during various panels. We could show that the QFT framework and ST paradigm are not mutually contradictory and can coexist together for the same consistent description of the QG. One can wonder about the issues of unitarity, causality, stability, non-locality or non-commutativity in QG or the true nature of degrees of freedom present there. For example, we think that all above adjectives apply to the true QG theory. Moreover, we believe that it is possible to find a description in which QG is as a QFT perturbative, conformal [307–309], UV-finite [173, 310–312] and

singularity-free [313] theory and in which spacetime still keeps its structure as a continuum. Additionally, in such a QG theory the S-matrix approach is valid, but we also learn things from general quantum correlators (based on Panel 7). This framework later can be applied to look for consistent description of quantum cosmology and quantum BHs.

In conclusion, we think that the future of QG can be bright, especially if new experiments or observations will bring some exciting news. As scientists working in a QG theory research, we need more discussion (like this was greatly promoted and reinforced at the workshop) and a resulting convergence of many different approaches towards QG that were created in the past. They really do not need to fight with each other (and this is very bad for the QG community)! Now we shall try and check whether some of them can be united. At the end, nature just realizes one successful approach to QG.

### 3.20 Sumati Surya:
### *The importance of being Lorentzian*

*I would like to dedicate this to the memory of Jurek Lewandowski*
*who actively participated in this meeting and was my co-panelist, despite his failing health.*

The meeting was characterised by intense discussion and debate on a set of issues which were interesting on several levels, ranging from scientific to sociological with some essential points of divergence. I highlight some important points of divergence.

**Discreteness versus the continuum (Panel 6):** Discreteness comes in many forms in QG, ranging from the area spectrum in LQG [314], to CDTs [9] and spin foams [315,316] where it is used as a regulator, and finally CST [15,317] where it is considered to be fundamental (see [318,319] for Riemann's attitude to discreteness). Continuum-based approaches adhere to an absolute continuum with strong and possibly unphysical requirements on differentiabilty [320]. The difference is exacerbated by the Lorentzian path integral "coming into being" of spacetime [9,315,317], in contrast to a stringent initial-value-based Riemannian spatial formulation.

**Non-locality:** How does one go from local field equations, exemplified by the standard phrasing of QFTs, to non-locality in QG? The dichotomy stems in part from what are deemed to be immutable, key features of QFT and the role of "local" differential operators in the standard mode decomposition of the quantum field. However, non-locality is evident in the very structure of the quantum field vacuum and explicit in the algebraic QFT perspective [321]. It suggests that a diffeomorphism invariant theory, with no (classical) local observables, should also be non-local, without necessarily violating causality. Another dichotomy presented itself in interpreting fluctuating lightcones. There is no fundamental dichotomy in the non-perturbative approaches, but in background dependent perturbative approaches there can be causality violation with respect to a "background causal structure".

**Gravitons and the S-matrix:** There were questions (including in Panel 7 and Panel 8) on why gravitons are not explicitly present in several non-perturbative approaches. This is related to the validity of a perturbative (possibly effective) QFT of gravity. It is, however, far from clear that gravitons are a *fact* of nature. LIGO confirms that there are *classical* GWs, but there is no known principle of physics which makes a quantum version of *these* waves an imperative. We do not quantise water waves but instead replace the macroscopic fluid with an underlying quantum molecular description. Not every classical aspect of a theory has an explicit and simplistic, fundamental/non-perturbative description as a quantum state. If graviton-like excitations can be constructed in non-perturbative theories of QG, either as coherent or even distributional states [322], they may bear little resemblance to the fundamental states of the theory. The idea of the S-matrix also came up as essential and immutable, and again, even in gauge theories, it is clear that there are issues with this — see [323] for a discussion.

**Direction in CST:** Coarse graining and renormalisation methods are natural to CST and it is a matter of interest to see how these can be carefully implemented [324, 325]. Interesting questions that arose during the meeting included the role of ASQG in a Lorentzian, fundamentally discrete theory like CST. Discussions indicated that these ideas are not necessarily in contradiction with discreteness and that there is room to understand the causal set partition function using effective actions and the RG.

**Vision:** I think debate and discussion is essential, but so too is explicitly enabling continued collaborations and discussions on the key issues. I think that QG is enriched as a field by such discussions and varied points of view and the understanding gained from different approaches. Sometimes we learn what QG is not and sometimes we feel we may have a hint at what it is. The workshop was very intense and very demanding in its ability to draw us all into engaged (sometime enraged!) conversation and I would love to see more of these over the coming years.

### 3.21 Arkady Tseytlin:
### *String theory and quantum gravity*

Regardless of whether ST is directly relevant to the description of our physical world, it undoubtedly provides a consistent model for QG. ST enables the systematic computation of UV-finite graviton scattering amplitudes, allowing for the unambiguous determination of coefficients for relevant terms in the low-energy effective action without requiring additional conjectures or assumptions.

For instance, within the framework of the AdS/CFT duality, the well-defined $1/N$ expansion of correlation functions in super Yang-Mills theory translates into a UV-finite string loop expansion in AdS space.

The standard approach in ST, based on summing over worldsheet surfaces or using string field theory, is primarily limited to perturbative expansions in the number of string loops, and thus, in the number of graviton loops. Understanding how to resum contributions from graviton loops remains a central challenge in QG. Recent advances leveraging AdS/CFT duality offer a method for computing graviton scattering amplitudes beyond the expansion in string coupling (see, e.g., [326, 327]). This approach utilizes localization techniques to exactly determine certain correlators of stress tensors in supersymmetric $SU(N)$ gauge theory with respect to both the gauge coupling $g_{ym}$ and $N$. The AdS/CFT correspondence then relates these correlators to graviton scattering amplitudes in AdS space, or in the flat-space limit. This duality provides a novel and potentially crucial perspective on the graviton S-matrix by connecting it to correlators in the boundary conformal field theory (CFT).

Another promising approach to resumming string loop perturbation theory is based on its connection to 11-dimensional M-theory, where quantum M2-branes are expected to play a central role. A major open question is whether advances in understanding M-theory could offer significant insights into non-perturbative QG. Recent evidence suggests that the quantization of M2-branes is consistent and offers a new pathway to accessing string loop corrections [328, 329]. There are indications that a hidden symmetry in M2-brane theory might imply that the theory remains well-defined beyond the one-loop order, despite its formal power-counting non-renormalizability. That may hold an important clue of how to define formally non-renormalizable theories in higher dimensions by analogy with integrable $T\bar{T}$ deformed models in two dimensions with possible implications also for QG models.

### 3.22 Neil Turok:
#### *Time to leave old silos and be guided by nature*

QG is an exciting and important frontier of basic physics. It must describe both the smallest and the largest scales in the universe: From the Planck scale to BHs and the dS horizon. It is uniquely difficult: Every approach so far pursued has serious shortcomings, so our best hope for progress is to encourage fresh, foundational thinking. The workshop was a wonderful opportunity for cross-fertilisation between approaches. The organizers created a friendly and respectful environment allowing plenty of time for discussions. This was the ideal way to sew the seeds for future, silo-busting collaborations.

Donoghue reminded us that physics is already remarkably unified. In flat spacetime and at energies below the Planck scale, perturbative quantum-gravitational effects can be reliably calculated using EFT. Since four-derivative gravity is renormalizable, it might yet provide a consistent framework at higher energies. Anselmi's new work, on removing ghosts systematically, order by order in perturbation theory, is a fascinating development. Is there an associated symmetry, analogous to gauge symmetry, or a superselection rule which restricts the Hilbert space to a positive subspace, non-perturbatively? The possibility of a UV fixed point, perhaps requiring specific matter, is extremely attractive, although the arguments presented so far seem tied to Euclidean methods and perturbation theory about flat space. Can they be extended to resolve singularities like the big bang, or inside BHs?

The most interesting questions about QG are non-perturbative. In the context of Euclidean QG, Percacci outlined a fascinating conundrum: What does a free scalar field contribute to the cosmological constant? Loll presented lattice simulations of Euclideanized "causal triangulations", finding tantalising evidence for a dS-like spacetime. Can such calculations shed light on Percacci's puzzle? Dowker presented causal sets as a physical basis for QG, along with intriguing arguments in favour of a continuum limit. One would like to see GR emerge more clearly: Do the discretized models predict GWs or BHs? Or cosmological spacetimes anything like ours? The tension between spacetime, pictured as a smooth manifold, and Heisenberg's uncertainty relation requiring it to fluctuate violently at short distances, continues to cry out for a clear resolution.

Our generation is fortunate to have so much more guidance than our predecessors. Observations and experiments have revealed a surprisingly simple, although deeply puzzling, universe on both the largest accessible scales and the smallest. The cosmos's regular, predictable nature encourages us to search for a unified theoretical description. Likewise, its greatest paradoxes, like the big bang singularity in our past and the strange, vacuous future which seems to lie ahead, provide some of the best clues we have about QG.

One of the most surprising aspects of the workshop, for me, was how few people are willing to question cosmic inflation. To accept such a contrived, ad hoc scenario seems to me to be blind to QG's full potential as well as its best chances of being observationally tested. Eternal inflation created a "measure" problem which was never resolved, and inflation's "smoking gun" signal, long wavelength tensor modes, hasn't shown up. The upper bound modes is set to fall dramatically over the coming decade. If it does, inflation will become less and less plausible. A CPT-symmetric, "mirror" universe [330] provides an alternative. Within this context, gravitational entropy, calculated à la Hawking, provides a well-defined, albeit semiclassical, statistical measure on the set of possible spacetimes [331,332]. The measure favours flatness, homogeneity and isotropy without any need for inflation, and suggests an explanation for the small positive cosmological constant. Resolving the big bang singularity with conformal symmetry likewise offers precise, testable predictions for the large-scale primordial density perturbations [333]. A minimal explanation for the dark matter [334], a new way to cancel the vacuum energy and trace anomalies [335], and an explanation for why there are three generations of fermions are other attractive features.

### 3.23 Thomas Van Riet:
### *Is the criticism on string theory justified?*

The workshops at the Nordita program provided an excellent opportunity for researchers on QG to step out of their echo chamber. I was able to learn more in-depth the reasons for the criticism on ST and in what follows I will address them based on the interactions during the various panel discussions (and coffee breaks) and end with a suggestion on how the QG community can continue in a productive and collaborative spirit.

**ST has no predictive power.** This is false since (weakly coupled) ST makes very clear predictions at the string scale: We see strings. Instead, the statement probably alludes to the difficulties ST has in making concrete predictions at measurable energy scales. I believe this is a trivial consequence from EFT and will be shared with all QG alternatives: If we agree that QG boils down to an EFT at low energies, then EFT lore implies that QG corrections to GR and the SM are suppressed by a high scale. One has to be lucky then for QG phenomena to be measured at low energy densities. Axions might provide such an opportunity.

**ST has a landscape.** This is true but that is not unscientific. It is a simple consequence of taking a theory in the UV and run it to the IR. Even the SM (which we all agree is a scientific theory) has this property. When the SM is coupled to (classical) gravity it allows for a gigantic landscape of lower-dimensional vacua [179], and around each vacuum it describes a different EFT. Only at the higher energy scale could one recover the 4d structure of the UV theory with its "unique SM couplings". Similarly ST is unique in the UV to the extend it has no tunable couplings at all. Yet it has, like almost all physics theories, a landscape of ground states where the fluctuations around it describe EFTs with different couplings. Interestingly, the swampland program [336] suggests that couplings of the various EFTs obtained from QG obey non-trivial patterns and this opens the door to a mild form of predictive power.

**ST might not provide sufficient non-locality to solve the information problem.** This was suggested in various panel discussions and this could be a justified criticism. ST has shown great levels of non-locality through the holographic correspondence, but is far from clear this level of non-locality suffices. If not, ST is not complete.

So how do we weigh alternatives against each other? I believe this is the core question and I hope the Nordita program was a first step to find consensus among the different approaches. We are in need of "wish list" for QG, independent of how far one approach has progressed to be able to address the wishes. I like to end this personal statement by making a few wishes.

(i) A theory of QG should allow one to compute graviton scattering without cut-offs that introduce unknown Wilson coefficients. In other words, a QG should allow us to derive Wilson coefficients for operators that are added to the SM coupled to gravity. I believe we have to accept these coefficients can depend on the choice of vacuum, but there is a good chance that they are unique once one identifies the vacuum that has the SM as a renormalisable sector.

(ii) A QG should address what is inside a BH and what constitutes the entropy of a BH (the microstates). It furthermore should tell us how information is contained in Hawking radiation, despite that radiation being unobservable in practice.

(iii) A QG theory should tell us about the early universe and what replaces the classical Big Bang singularity.

(iv) A QG theory should be a theory of spacetime and answer whether spacetime is fundamental or emergent.

As you can see this wish list is largely independent of one's ability to measure directly QG but contains statements that all have to do with internal logical consistency of a theory. Internal logical consistency is step one in all of science. Step two is the interplay with observations. It is completely fine for humanity to deal with step one without step two because of technical obstacles. If step one would not be constraining QG would not be physics. But step one has shown to be so strongly constraining that QG is definitely part of solid and exciting science.

### 3.24 Richard P. Woodard:
*Aha!*

My chief take-away from the meeting is finally understanding, in terms of conventional QFT, what the ASQG people are claiming. The claim is that they have guessed an effective action for QG, and that they can reconstruct the classical action whose quantization it represents. They claim that there are three fixed points, none of which corresponds to quantizing GR. One of these fixed points represents the $R + R^2 + C^2$ theory of Stelle, which they dismiss on account of its ghosts. The fixed point for which they hold out hope would correspond to an invariant, metric-based and non-local classical theory of gravity. They believe that it does not suffer from the Ostrogradskian instability because its non-locality takes the form of entire functions of the derivative operator, which induce no extra poles in Euclidean momentum space. I very much doubt that this theory is acceptable because making the assumption that the temporal Fourier transform exists amounts to assuming away precisely the pathological time dependence one encounters with the Ostrogradskian instability.

## 4 Individual thoughts in the light of discussions:
*Quantum gravity phenomenology*

### 4.1 Robert Brandenberger:
*Demise of the effective field theory approach to quantum gravity*

I was fortunate to be able to attend week 4 of the 2024 Nordita QG program. Here are some thoughts based on my interactions with other participants, from the point of view of a theoretical cosmologist (I do not consider myself to be a QG expert).

An important point I tried to stress throughout the week is that point particle EFTs for matter and gravity are non-unitary in an expanding background [34, 105, 337]. Since an EFT requires a UV cutoff at a fixed physical scale, modes need to be created continuously in an expanding universe. As long as the modes which are continuously created do not exit the Hubble horizon, one may argue that this problem is irrelevant for phenomenological issues (cosmological observations). The problem, however, is fatal for standard inflationary models [106]. This has implications for issues discussed in a number of the panels. It appears to me that QG cannot be described in a QFT setting (Panel 1), and must be (Panel 8) based on a non-perturbative framework. A non-perturbative approach will then indicate what classes of low energy EFTs are viable and which are in the "swamp". This is the goal of the "swampland program" (Panel 3), which I view as a very important and promising research direction.

These above considerations have implications for QG phenomenology, the topic of week 4 of the program. I consider claims of signatures of QG in cosmological observations as very suspect if they are based on EFT computations, as done e.g. in loop quantum cosmology (LQC). On the other hand, searching for imprints of non-perturbative approaches to QG in cosmological observations is very promising (Panel 10).

There are a number of interesting approaches to QG based on non-perturbative methods. In particular with applications to cosmology in mind, it appears to me that an approach to QG which considers matter and gravity in a unified framework will be most promising, and superstring theory to my knowledge provides the best starting point. However, we are still missing a well-established non-perturbative definition of ST (Panel 4). Note that compared to the swampland of possible EFTs, superstring theory is very restrictive. This is an important point missed in most of the attacks on ST.

As an example of how QG could lead to observable signatures, I mentioned our approach on matrix cosmology [338]. We start from a well-defined quantum mechanical matrix action (the BFSS action [339]) which was proposed as a non-perturbative definition of superstring theory. Considering a high temperature state of this matrix model, it is possible to obtain an emergent space-time [340, 341] and an emergent early universe cosmology which is very similar to the "string gas cosmology" toy model of [342]. Thermal fluctuations in this state yield curvature perturbations and GWs on cosmological scales with scale-invariant spectra. The distinctive prediction is a slight blue tilt of the spectrum of GWs [343], a prediction which differs from what is obtained in the standard inflationary paradigm. I view this as demonstration that QG can make specific predictions for interesting cosmological observables.

### 4.2 Anne-Christine Davis:
*What are we missing?*

For the past 25 years we have had a very big hint that something is missing in the standard model of cosmology. Originally, supernova data showed the universe was undergoing a period of accelerated expansion today. This original observation has been cemented by CMB data from the Planck satellite. It is now accepted and we try to explain this as either being due to a cosmological constant or to some modified gravity theory, usually involving an extra scalar field. Neither explanation is satisfactory in my opinion, although I have worked on the latter for a number of years. A cosmological constant requires a value that is 120 orders of magnitude below the natural value in GR, whilst modified gravity theories have so far failed to provide a satisfactory answer. On the other hand, the two pillar stones of 20th century physics, namely quantum theory and GR, have explained all of the known 20th century physics.

However, try as we might gravity has eluded quantisation. Is it time for a new theory to step forward? It would have to reduce to GR in a limit since the solar system is well described by GR. Is quantum theory actually correct or is there something beyond? These are puzzles I cannot answer but suspect the so-called cosmological constant is a hint of new physics. Of course any new theory would have to contain GR and quantum theory in a similar way that Einstein gravity reduces to Newtonian gravity in certain limits.

- In discussion, it appears there are developments in ST that I was unaware of and may go some way towards offering explanations. It is possible that SM physics could come out of ST.

- In Panel 9 and Panel 10, discussions have opened up possibilities for further exploration. I emphasized the cosmological constant as a hint towards new physics, others explained their viewpoints. As for testing QG, there are a number of laboratory experiments that are now so accurate that deviations could be explained by "new physics". These include atomic interferometry experiments and Casimir force experiments. There are other experiments specifically designed to test QG, but I'm a great believer in looking at existing experiments and seeing the possibilities there.

- QG is an emerging field and one that will become increasingly important as time goes on. I suspect it requires thinking "outside the box" in a more radical way than has been done so far. However, it is highly unlikely that the two tenements of 20th century physics are completely separate. We must persevere.

## 4.3 Astrid Eichhorn:
### *Quantum gravity phenomenology — Towards comprehensive tests of quantum gravity at all scales*

A connection of QG to observations is critical in order to achieve a true breakthrough in our understanding of quantum properties of spacetime. Current research developing observational tests of QG proceeds mostly along two largely disconnected lines, namely developing tests of smoking-gun signatures or developing IR consistency tests. I will argue that a more connected and coherent effort is required to make maximum use of all possible observational constraints.

There are two types of smoking-gun signatures, distinguished by the underlying assumption about the scale of QG.

If the scale of QG is the Planck scale, then "smoking-gun" signatures need a lever arm that enhances the effect, because experiments do not (yet) reach the Planck scale. An example is the suggestion that QG modifies dispersion relations and these effects accumulate, e.g., in the propagation of photons over astrophysical distances, with distance acting as a lever arm [344], see [145] for a review. A second example is the idea that spin acts as a lever arm in BHs [141, 142], such that the horizon structure of near-extremal BHs is susceptible to Planck-scale QG corrections.

A challenge with this line of inquiry is that not all systems exhibit such lever arms. Thus, there is a danger of a "lamppost" effect: We search only for proposed QG effects that are accessible through lever arms, but these effects may not necessarily be the best motivated ones.

A less conservative approach to "smoking-gun" effects of QG is to conjecture that they occur at a scale much larger than the Planck length. Examples are the conjecture of a non-locality scale in causal sets [345], a large new-physics scale linked to the regularization of BH singularities [346] or the existence of large extra dimensions in ST. A challenge with this approach is that typically such a large separation between the Planck scale and a second, much lower energy scale, is difficult to achieve in a theory and most examples which are explored in phenomenology are based on ad-hoc assumptions about the second scale.

The idea behind IR consistency tests is different and relies on a general and deep principle about physics, namely the idea that microphysics determines macrophysics (and not vice-versa). As a consequence, free parameters of effective descriptions of a system are calculable in terms of the microphysics of the system. As an additional consequence, a given effective description may not have a viable, underlying microphysics and may thus not be an effective description applicable to nature. Both aspects are actively pursued in QG. For instance, the string-inspired swampland program [65] excludes EFTs that do not have a UV completion in ST, see [66–69] for reviews. Similarly, ASQG excludes some EFTs (see [347] for a dark-matter example) and constrains others (see [348, 349] for SM examples), see [292, 350, 351] for reviews.

Currently, QG phenomenology is rarely pursued by combining all possible lines of inquiry. It is to be expected that we are therefore missing important complementarity in different tests. For instance, what are the expected smoking-gun-effects *and* IR consistency tests of a Lorentz-violating theory? It may well be the case that a consistent theory which produces observable deviations in dispersion relations can be much more strongly constrained by IR consistency tests. This may in particular be true if a smoking-gun effect is tied to an RG relevant interaction, the size of which increases towards the IR, resulting in very strong deviations of IR observables

from experiment, see [116] for an example. Such synergy is missed, unless all possible ways to constrain QG by observations *at all scales* are accounted for in one comprehensive framework.

## 4.4 Giulia Gubitosi:
### *Following different trails in a bottom-up journey to quantum gravity*

A major advance in QG research has been marked by the transition from the epoch where QG was a matter of purely theoretical investigations, where scientists aimed at developing models for QST based on first principles, to the epoch were this kind of research is supported and complemented by efforts to link QG to observations [352].

In the past couple of decades, several observational frameworks were identified, where genuinely new physics effects related to QG could be observable. Some notable examples mentioned during the workshop were: Astrophysics (tests of spacetime symmetries) [145], tabletop experiments on quantum systems (tests of locality, tests of gravitational quantum physics) [117], GW astronomy (tests of large extra dimensions, singularity resolution) [353].

Despite the fast progress on the phenomenological side, how to link full-fledged QG theories and our observable world is still a matter of heated discussion. To explicitly work out in a rigorous way the low energy/classical spacetime limits is not possible due to the complicated structure of QG theories. In the best case, a number of (sometimes radical) simplifying assumptions are required, thereby weakening the link between the fundamental theories and their phenomenological predictions.

For this reason, several communities take an alternative, bottom-up, approach, which starts from currently established theories and builds up from them, in the hope of capturing residual QG effects that could be present in a mesoscopic regime, intermediate between the classical spacetime/low energy regime of current theories and the full QST/Planck-scale regime [354]. During the workshop, several possibilities were discussed in this respect (see discussions in Panel 10 and Panel 11). EFT and, closely related, higher derivative gravity rely on the assumption that there is a separation between the (known) low energy quantum effects and the (unknown) high energy contributions. In the ASQG program the RG flow governs the IR/UV transition. Going beyond the QFT approach, possible modifications of special relativity are studied within a quantum groups approach or within the relative locality framework [355], departures from standard quantum mechanics are studied through modified dynamical evolution and the notion of quantum observers [356–358], and generalized spacetime geometries are studied by means of non-Riemannian structures [359].

While these approaches are often seen as competing, with the underlying assumption that just one of them will turn out to correctly describe the QST limits of QG, discussions in Panel 10 and Panel 11 raised the possibility that these are in fact paths that explore different intermediate regions between the known world and QG, all of which are relevant in the search for phenomenological constraints.

Understanding whether these regimes are related and characterizing the different corners of semiclassical physics that they describe would allow us to better assess current observational constraints by defining their framework of validity and to open different bottom-up paths available for reach by top-down approaches, therefore improving the connection between fundamental QG theories and testable new phenomena.

## 4.5 Lavinia Heisenberg:
### *Restricting the landscape of effective theories for gravity*

In the absence of a fundamental quantum theory of gravity, EFTs are indispensable tools for modelling gravitational phenomena and studying their observational implications. However, the vast degeneracy among these theories presents a significant challenge. To meaningfully

constrain the landscape of EFTs, it is essential to incorporate assumptions about the UV properties of gravity, and explore how fundamental quantum principles influence low-energy effective descriptions. One powerful approach is to evaluate whether prominent gravitational EFTs can be embedded within a broader QG framework, such as ST, through programs like the swampland initiative. This program identifies criteria that EFTs must satisfy to remain consistent with UV-complete theories of gravity, effectively excluding theories that fail these criteria. Additionally, theoretical constraints such as positivity bounds on scattering amplitudes provide critical insights into the physical viability of specific parameters within these EFTs. Another crucial aspect involves studying quantum corrections to classical gravitational Lagrangians. By examining whether these EFTs remain stable under loop quantum contributions, we can determine the robustness of their predictions and their consistency with fundamental quantum principles. These theoretical constraints, when combined with observational data, form a comprehensive strategy for narrowing down the expansive landscape of gravitational theories. Ultimately, this multi-faceted approach — merging theoretical restrictions from UV considerations and quantum consistency with empirical observations — holds the promise of refining our understanding of gravity and advancing us closer to a unified quantum theory of gravity.

### 4.6   Stefano Liberati:
### *The tumultuous life of quantum black holes*

Gravitational research is at a turning point, with data from GWs [360], very-long-baseline interferometry BH imaging [361], and cosmological observations allowing us to explore physics beyond GR. The QG phenomenology week reinforced the sense that we are at a crucial juncture, where theoretical ideas are finally intersecting with observational data. Particularly exciting is the collaboration between various fields — particle and gravitational physics, astrophysics, and cosmology — on a wide range of topics from tests of UV spacetime symmetries to probing BH geometries or quantum effects in the early universe.

A key focus has been on quantum BHs [362], which have shifted from speculative concepts motivated by the information loss problem to viable alternatives for astrophysical BHs observed through multi-messenger data. The term "quantum BH" broadly encompasses models modifying classical GR solutions due to semiclassical or QG effects. These objects, once remote from observational consideration, now lie at the center of new investigations, thanks to advances in our ability to probe them.

The QG capability to regularize BH singularities opens exciting possibilities. For example, it might lead to stable Planck-scale remnants linking primordial BHs to dark matter [363], while larger regular BHs were can be classified in a limited number of cases [147], with differing core structures leading to distinct solutions, not just in the interior but even near-horizon region outside the BH.

Another striking discovery is that inner horizons — characterizing several regular BHs but also Kerr/rotating BHs of GR — are generally unstable, both classically [364] and semiclassically [365–367]; even in non-stationary geometries where they are not also Cauchy horizons. This instability strongly suggests that BH cores will be generically different from those predicted by GR, and that trapped regions may be highly dynamical: A metastable phase of gravitational collapse rather than its true end-point.

It remains unclear what the final state of this instabilities will be, but it is intriguing that many regular BH geometries proposed so far could be smoothly deformed to describe horizonless ultra-compact objects in certain regions of their parameter space [368]. These objects, albeit hard to distinguish from BHs, could reveal their nature through unique features such as GW echoes [369, 370] or complex shadows [371]. Note however, that these ultra-compact objects, as well as extremal BHs [372], still suffer from several instabilities, such as ergoregion [373] and light ring instabilities [374]. Hence, the true endpoint of gravitational collapse,

once QG effects within trapping regions are considered, remains an open and exciting question.

An additional intriguing point is that the perturbative renormalizability of certain QFTs, like quadratic gravity or projectable Hořava theory, does not automatically resolve singularities (e.g. [375, 376]) while non-perturbative/discrete QG approaches seems to generically do so (e.g. [153, 377]). This naturally raises the question: What crucial feature enables a QG model to regularize singularities? At first glance, it seems that a fundamental discretization or "atomization" of spacetime might be necessary, but even in this context, our understanding remains incomplete.

In conclusion, QG phenomenology and quantum BHs lie at the frontier of theoretical physics. Solving challenges like the BH information paradox, the nature of singularities, and finding evidence for quantum BHs or ultra-compact objects could provide deep insights into the nature of reality. The sooner we shall embrace these challenges and dedicate ourselves to bridging the gap between QG and observations, the sooner we shall begin to uncover the missing pages of nature's book.

## 4.7 Paulo Moniz:
### *Quantum cosmology – No boundaries?*

Most of the open questions and still current challenges in QC can be found, for example, in specific chapters or sections in [100, 124, 378–380], from which I took some of the information herewith summarized. QC is still very much uncharted ground that aims to tackle the quantum description of the early universe.

In the QC workshop (as a speaker) and the two panels attended (as a panelist in Panel 9 and Panel 10), it became either pertinent or useful that it is of relevance [380] to cover the basics, discussing ideas and concepts of QC; summarize *what we know, what we think we know and we think we do not know*, always focused on "young", inquisitive minds, eager to embark, ask the "right" questions (rather than seeking immediate answers); see [100].

In particular, although work has been invested, more needs to be added into clearer routes regarding how

(i) Can QC become "observational"? See, e.g., in [124]. Specifically, can supersymmetric QC [378, 379] become "observational"? Would inflation and structure formation (Bunch-Davies vacuum?) be robustly predicted more generally? Concretely, beyond (see [124]) GR setting and a plain scalar field (inflation) associated?

(ii) Could torsion be found in any spacetime test and be relevant? Specifically, if hinting at some gravitino presence at the origin or suitable description of such a torsion feature [378, 379]. What would change in our understanding of nature?

The above paragraph summarises topics conveyed and employed to motivate the panels and audience.

Likewise, the discussion about inflation being not robust in EFT (in a curved spacetime) and advances in boundary conditions for the universe's wave function was pertinent. Discussing spacetime physics beyond metricity may also be of disclosure.

One aspect to consider is that any QC setting must lead towards a consistent QFTCS as an operational framework. This means making concrete predictions that can eventually be tested. This is made explicit in several papers from the late '90s and is described in chapter 6 of [100]. On the one hand, putting constraints on what a QC can present (because the QFTCS must meet observational unequivocal features): Either in GWs, hints of gravitons, polarization and deviations from GWs as indicated in gravity (classical) theories, as well as deviations from

classical CMB radiation features brought about by QFTCS, higher terms from perturbation from curvature and hence a hint from QC.

Although not covered in my talk, I think LQC is to be reminded about as well as stringy (quantum) cosmology and all possible consequences (e.g., pre Big Bang). All convey particular intrinsic predictions or suggestions of concrete imprints that may be (or may not) eventually found.

On a "personal" note, fractional calculus can provide hints regarding features of spacetime beyond the spacetime continuum; see in [100]. Likewise, the issue of spacetime WKB ruptures as QG corrections become dominant, leading to problems (e.g., non-unitary). One direction mostly avoided is to use quantum chaos methods within QC; see also in [100]. Last but not least, it was curious to perceive that the degeneracy issue in the universe's quantum wave function, as trying to be addressed by Picard-Lefschetz's recent methods, is somewhat related to supersymmetry (see in [100].). If either from the LHC or any advanced post-Planck mission, a hint or signature associated with supersymmetry could be identified, allowing us to consider supersymmetric QC viable, this would be a grand moment. It will be new physics, on the one hand. Still, it will also open up navigation upon some domain to chart upon which "technology", meaning concrete (mathematical and physical) framework, has been built and can be rational, i.e. scientifically tested [378, 379].

## 4.8 Olga Papadoulaki:
### *How to pinpoint horizons in quantum gravity*

My participation in Panel 12 was very illuminating in relation to how different communities that work in the realm of gravity regard BHs as well as what are the open problems that each community is trying to address. The discussions in Panel 12 ranged from the observational signatures of QG in astrophysical BHs to remnants, the distinction between horizonless gravitating solutions and typical BHs as well as the thermodynamics and stability of exotic gravitating objects.

There were many interesting points that came out of the interactions during the panel discussion. I would like to highlight the need for identifying observables that can distinguish between a typical BH (a gravitating solution that possess a singularity and a horizon) from gravitating horizonless objects, such as fuzzballs. For example could echoes or quasi-normal modes give us this information? Are there any special operators whose correlators could distinguish between the two?

Another very important point that was raised during Panel 12 was when one should expect the classical theory for the study of BHs to break down and quantum effects to start playing an important role and given this what type of modifications and at what level should be made to the classical theory.

Finally, in a theoretical level there are non-Schwarzschild BHs that arise from (super)gravity theories that appear to be stable. There are many subtleties regarding the thermodynamic properties of such solutions that should be elucidated.

From the various questions mentioned above, I am very eager and excited to explore potential observables that can distinguish between horizonless gravitating objects and BHs. This question has both theoretical and phenomenological implications as was made clear during the panel discussion Panel 12. Specifically, I would like to ask this question in the context of two dimensional ST and matrix quantum mechanics [156]. The reason is that as has been shown in [157], there are two regions of the phase space of the possible solutions of Liouville ST coupled to a matter boson where BH like physics is present. It is possible then that one of these regions corresponds to a horizonless gravitating object and the other one to a typical BH with a horizon. Due to the fact that the matrix model description allows for analytical

computations this is a framework where this question has the potential to be answered in a closed analytical form.

My background is in ST and holography. I participated in the workshop on QG phenomenology. Phenomenology is far away from my expertise thus this participation gave me the opportunity to gain a different perspective as to what are the important questions that need to be answered as well as how different communities think about BHs and QG as a whole. I believed I gained many new insights that I will be able to use to shape the future directions of my research.

### 4.9 Mairi Sakellariadou:
*Quantum gravity in the golden age of cosmology*

QG is the still awaited theory describing physical phenomena at the cross road of GR and QFT. There exist several proposals, however it is still unclear whether any of those has any connection with "the" QG theory, the main reason being a lack of experimental data. Several tabletop analogue experiments attempt to mimic the physical situation where GR and QFT are simultaneously applicable, namely to describe a particle with Compton wavelength equal to its Schwarzschild radius. However, a laboratory experiment aimed at testing a QG proposal requires energies up to $10^{19}$GeV and such an energy level is above the reach of any laboratory we can dream of. The only way out is to use the information we can retrieve from an experiment going on for almost 14 billion years, namely the evolution of our own universe. Planck energies were reached in the very early universe and footprints of that era can be found in the universe within which we live. Today, we receive a plethora of astrophysical data from different sources and different physical processes: We live the golden age of cosmology. Affinities between QG and cosmology imply that on the one hand cosmology provides a test of a quantum theory of spacetime geometry and on the other hand QG can solve open problems in cosmology or even explain the emergence of our universe. There is however a catch: Raw data are analysed within a particular physical framework, which implies that at the end of the day we do not have an agnostic test but we just perform a consistency check. One has to be particularly careful to avoid any bias in the interpretation caused by the particular physical framework we use to analyse the data.

My views about (some of) the panels are as follows:

- QFT is not sufficient to formulate a theory of QG and we have an important handicap: We may not yet have the appropriate mathematical tools to formulate such a theory.

- To understand geometry at Planck scales, we should abandon the notions we are familiar with: Spacetime becomes a wildly non-commutative manifold. It is important to note that quantum mechanics is intrinsically non-commutative. This automatically implies that at such energy scales spacetime is discrete.

- The string paradigm is a rich mathematical theory with a serious drawback if one is trying to promote it to the status of it is a physical theory, namely the landscape problem.

- QG is definitely non-perturbative, even though perturbative approaches may give some helpful insight.

- EFT and modified gravity proposals can provide some information but if one cannot motivate them from any fundamental approach they remain phenomenological working examples.

The road is still long to get to a theory of QG. All approaches are useful and may shed some light on the correct theory of QG, provided one is carefully keeping in mind which assumptions, simplifications and choice of framework are made. The interplay between sophisticated mathematical tools and observational data is an important asset.

### 4.10 Alberto Salvio:
#### *Exploring the landscape of phenomenologically viable quantum gravities?*

Perturbative approaches to QG require giving up or extending some of the pillars of relativistic QFT [381]. Then four approaches come to mind:

**Giving up the field-theory structure.** Relativistic QFT emerges as the only possible theory with a particle spectrum unifying the standard principles of quantum mechanics and special relativity. Thus, one may attempt to relax the hypothesis of a particle spectrum. This is the approach pursued by ST. ST phenomenology is currently trying to understand whether ST can be compatible with all observations and experimental data we have.

**Giving up relativity.** Introducing a privileged reference frame could render gravity renormalizable without modifying any other principle of QFT, as proposed by Hořava [382]. Work is ongoing to determine whether this is possible in a fully realistic setup.

**Extending quantum mechanics.** Adding terms quadratic in the curvature renders Einstein gravity renormalizable (quadratic gravity), but an indefinite inner product on the Hilbert space has to be introduced [37]. This can be made compatible with a consistent probabilistic system (and thus unitarity) if quantum mechanics is extended at high energies [383, 384]. Indeed, the indefinite inner product is not the one to be used in the Born rule (to compute quantum probabilities): A physical interpretation of probabilities shows that a positive inner product (which exists) should play that role [385, 386] and one finds a multi-inner-product theory.

**Non-perturbatively, ASQG** (the existence of a predictive UV fixed point, or more precisely a finite-dimensional critical surface) [185] can leave intact all the standard assumptions of relativistic QFT, but it is computationally challenging.

It is important to keep working on all possibilities that have a chance to be phenomenologically viable. The non-perturbative formulation of quadratic gravity [384, 387] presented in the program holds both in the perturbative and non-perturbative regime. In the perturbative one the high-energy scale where quantum mechanics is extended could be reached by future observations; non-perturbatively it might be possible that standard quantum mechanics keep holding at all observationally reachable energies within ASQG [388, 389]. Establishing the existence of a continuum limit is a challenge, just like for lattice QCD, for future research that was emphasized in the discussion session after my talk. This is related to other QG approaches such as CDT. But in quadratic gravity the conformal factor problem is avoided.

Predictivity is very important, but being consistent with observations is even more so. What are the viable QG approaches (including viable non-gravitational interactions and matter)? Possible arenas to constrain QG approaches are the physics of the early universe and that of BHs. In the context of quadratic gravity natural inflaton candidates are the scalaron (a.k.a. Starobinsky inflaton) and the Higgs, which have been explored in realistic setups in [53, 306, 390–392]; moreover, several studies of ultracompact objects, with or without horizons, have been conducted [130, 393, 394], but questions regarding their stability and the endpoint of a physical gravitational collapse remain so far unanswered.

### 4.11 Kellogg Stelle:
#### *Key issues linking the different approaches to quantum gravity*

The panel discussions at the workshop on QG phenomenology focussed on a number of central issues in the search for a quantum theory of gravity and also on the concordance between some

of the main lines of attack on this unsolved area of fundamental physics. A key relation to be settled is that between the ST approach and the various aspects of ordinary particle QFT. EFT can provide a framework for a bridge between fundamental theory at an underlying ultra-microscopic scale and more phenomenologically tractable treatments at scales more relevant to observation. One hope for such a bridge between scales could be the ASQG program, which might encode QG phenomena in a field theory language relevant to sub-string-scale physics which could be applicable to questions in the early universe and to BH physics.

One characteristic question that arises in QG investigations concerns the role of fourth-order in derivatives quadratic-curvature counterterm structures which appear in essentially all QG approaches, including ST. If included in the initial action, these quadratic curvature terms yield a UV stable perturbative expansion, but this at the cost of new unstable "ghost" modes not present in the classical second-order theory. The ghost mode poles could, however, develop imaginary parts indicating instability and decay, thus at least pushing off the scale of reckoning to values where effective theory has to give way to a true ultra-microscopic theory. If so, what is the impact of such higher derivatives in important questions like the early universe? One attractive possibility is to generate constraints on the initial cosmological boundary conditions, possibly fulfilling Penrose's Weyl curvature hypothesis for a low-entropy beginning to the universe. Another question is the consequences for BH solutions. The quadratic curvature terms, generating effective field equations fourth order in derivatives, yield additional BH solution families at low mass values. What impact do such solutions have both in astrophysics and in the early universe, or indeed on the BH evaporation question?

More general issues to be confronted in the concordance between the different QG lines of attack are encoded in the "swampland program", which proposes to determine which classical or semiclassical approaches to QG are capable of ever being incorporated into a UV complete quantum theory. These questions also reflect back on some of the most glaring unsolved issues such as the role or even existence of dark energy.

# 5 Conclusions by the organizers: The importance of cross-fertilization of ideas and approaches in quantum gravity

**Extensive discussions as QG-catalysts.** Formulating a consistent theory of QG is one of the most outstanding open problems in theoretical physics. Several proposals for such a theory exist, each facing its own challenges and making progress in different directions. We organizers believe that sharing ideas and methodologies can act as a catalyst for progress in the field. With the Nordita Scientific Program *"Quantum Gravity: From gravitational effective field theories to ultraviolet complete approaches"* and with this contribution, we experimented with something genuinely new: We brought together most approaches to QG with the scope of *extensively* discussing common grounds and identifying sources of disagreement. To this end, during the three weeks of workshops, we had 15 open discussions and 12 panels, totaling about 40 hours of heated debates and exchange of ideas. In addition, there were about 30 hours of talks, where individual topics were introduced pedagogically to set the stage for the daily discussions. This setup generated a platform for learning about achievements and drawbacks of other approaches, exchanging ideas, and ultimately driving progress in QG research. The atmosphere was one of respect and openness — despite several disagreements, participants actually tried to listen to praises and criticisms from other approaches, and shared thoughts with an open mind. Many participants reported that the discussions were very eye-opening, and that they gained many new ideas that they are looking forward to developing.

**Overall thoughts from the organizers.** Having attended all lectures and discussions during the four-week program, we organizers can summarize our impressions in one sentence: The self-sustained, heated discussions of the program clearly demonstrated that the QG community needs more such events with *extensive* discussions. To us, the most striking aspect highlighted by the program was that different approaches to QG even disagreed on *which questions* are important to ask and address: There was disagreement on whether EFT should be considered a common ground, on whether gravity is a gauge theory, and on whether one should recover a graviton at low energy. Based on this, the focus questions of each approach are different, ranging from the consistent form of scattering amplitudes to spacetime emergence and relational observables. Generally, there were two big lines of thought: String and quantum field theorists agreed on the role of EFT as a common ground, while theorists supporting discreteness as a fundamental feature of QG were somewhat skeptical about the latter points. Finally, we noted a significant difference between theory- and phenomenology-oriented discussions: The former (Panels 1-8) saw strong disagreements and heated discussions, but also a lot of clarification and knowledge gain, the latter (Panels 9-12) stepped from widely-spread agreements on GR-induced physics to wild discussions on the many possibilities that one faces when leaving the classical theory. All in all, understanding and constraining the low-energy predictions of different approaches is crucial to restrict the range of possibilities. This should then be the seed for phenomenological studies, which everyone agreed should be strongly driven by fundamental principles in QG. Which principles govern the physics of our Universe at the fundamental level is yet a question to unravel.

**Concluding remarks.** We hope that this Nordita program has acted as an effective source of ideas and constructive debates that will benefit not only the participants, but also any researcher who will read this contribution or watch the recordings. We strongly believe that meetings like this will ultimately pave the way to new insights and progress in QG. In a time where it is difficult to even keep up with the developments in one's own field, let alone in neighboring fields, this cross-fertilization of ideas and approaches via *intense discussions* rather than an excess of talks is more important than ever. It is clear to us that events of this kind should be organized more often, and we hope that our efforts can serve as an inspiration for the future.

# Acknowledgments

Luca Buoninfante, Benjamin Knorr, K. Sravan Kumar and Alessia Platania would like to thank Nordita for sponsoring the Nordita program. Anne-Christine Davis would like to thank all of her collaborators over the years who've worked with her on modified gravity, and in particular Philippe Brax. Daniele Oriti thanks Jan Ambjørn, Bianca Dittrich, Alessia Platania, Roberto Percacci, all members of the panels, all participants to the workshop, and especially Ivano Basile, for stimulating and instructive discussions.

**Author contributions** The summaries of the panel discussions were written by Luca Buoninfante (Panels 2, 3, 9, 11), Benjamin Knorr (Panels 6, 7, 8, 10), Sravan Kumar (Panels 9, 10, 12), and Alessia Platania (Panels 1, 4, 5, 12). The introduction and conclusions were jointly written by Luca Buoninfante, Benjamin Knorr, K. Sravan Kumar, and Alessia Platania. All remaining authors were responsible for the individual subsection bearing their name, and their views and opinions are not necessarily shared by the other authors. All speakers and panelists were given the opportunity to give feedback on the panel summaries. The style and formatting in TeX were handled by Benjamin Knorr.

**Funding information** The program was funded by Nordita. Nordita is supported in part by NordForsk. The authors' funding information is as follows.

- The work of N. Emil J. Bjerrum-Bohr is supported by DFF grant 1026-00077B and in part by the Carlsberg Foundation.

- Luca Buoninfante is financially supported by the European Union's Horizon 2020 research and innovation programme under the Marie Skłodowska-Curie Actions (grant agreement ID: 101106345-NLQG).

- Mariana Carrillo González' work is supported by the Imperial College Research Fellowship.

- The research of Bianca Dittrich is supported by Perimeter Institute. Research at Perimeter Institute is supported in part by the Government of Canada through the Department of Innovation, Science and Economic Development Canada and by the Province of Ontario through the Ministry of Colleges and Universities.

- Paolo Di Vecchia's research is partially supported by the Knut and Alice Wallenberg Foundation under grant KAW 2018.0116.

- The work of John F. Donoghue is supported in part by the US National Science Foundation under grant NSF-PHY-21-12800.

- The research of Gia Dvali is supported in part by the European Research Council Gravities Horizon Grant AO number: 850 173-6, by the Deutsche Forschungsgemeinschaft (DFG, German Research Foundation) under Germany's Excellence Strategy - EXC-2111 - 390814868, and Germany's Excellence Strategy under Excellence Cluster Origins. Disclaimer: Funded by the European Union. Views and opinions expressed are however those of the authors only and do not necessarily reflect those of the European Union or European Research Council. Neither the European Union nor the granting authority can be held responsible for them.

- Astrid Eichhorn is supported by a grant from Villum Fonden (29405).

- The work of Steven B. Giddings is supported in part by the Heising-Simons Foundation under grants #2021-2819, #2024-5307, and by the U.S. Department of Energy, Office of Science, under Award Number DE-SC0011702.

- The work of Renata Kallosh is supported by the US National Science Foundation grant PHY-2310429.

- Benjamin Knorr was partially supported by Nordita.

- K. Sravan Kumar is supported by the Royal Society's Newton International Fellowship.

- Paulo Moniz acknowledges the FCT grant UID-B-MAT/00212/2020 at CMA-UBI plus the COST Action CA23130 (Bridging high and low energies in search of quantum gravity (BridgeQG)).

- Daniele Oriti acknowledges financial support through the Grant PR28/23 ATR2023-145735 funded by MCIN /AEI /10.13039/501100011033, through the Grant FR62421-RQ-46152 funded by the J. Templeton Foundation, and through the Core Membership funds by the MCQST.

- Jan M. Pawlowski is funded by the Deutsche Forschungsgemeinschaft (DFG, German Research Foundation) under Germany's Excellence Strategy EXC 2181/1 - 390900948 (the Heidelberg STRUCTURES Excellence Cluster) and under the Collaborative Research Centre SFB 1225 (ISOQUANT).

- The research of Alessia Platania is supported by a research grant (VIL60819) from VIL-LUM FONDEN.

- Sumati Surya is supported by SERB Metrics Grant Number MTR/2023/000831.

- Arkady Tseytlin is supported in part by the STFC Consolidated Grant ST/T000791/1.

- The research of Thomas Van Riet is in part supported by the Odysseus grant GCD-D5133-G0H9318N of FWO-Vlaanderen.

- Richard P. Woodard is supported by the National Science Foundation under the grant PHY-2207514.

## List of acronyms

**AdS**      anti-de Sitter

**AdS/CFT**  anti-de Sitter/conformal field theory

**ASQG**     asymptotically safe quantum gravity

**BH**       black hole

**CDT**      causal dynamical triangulations

**CST**      causal set theory

**CFT**      conformal field theory

**CMB**      cosmic microwave background

**dS**       de Sitter

**EFT**      effective field theory

**GR**       General Relativity

**GW**       gravitational wave

**GFT**      group field theory

**IR**       infrared

**$\Lambda$CDM**    Lambda cold dark matter

**LHC**      Large Hadron Collider

**LQC**      loop quantum cosmology

**LQG**      loop quantum gravity

**MAG**      metric-affine gravity

**MOND**     Modified Newtonian Dynamics

**QCD**      quantum chromodynamics

**QC**       quantum cosmology

**QED**      quantum electrodynamics

**QFT**      quantum field theory

**QFTCS**    quantum field theory on curved spacetime

**QG**    quantum gravity

**QM**    quantum mechanics

**QST**    quantum spacetime

**RG**    renormalization group

**SM**    Standard Model of Particle Physics

**ST**    string theory

**TCC**    trans-Planckian censorship conjecture

**UV**    ultraviolet

**WGC**    weak gravity conjecture

**WKB**    Wentzel–Kramers–Brillouin

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
