# Peer review of "Visions in Quantum Gravity"

_SciPost Physics Community Reports, doi:SciPost Phys. Comm. Rep. 11 (2025)_

## Round 2 · Referee Report · Anonymous (Referee 1) · 2025-5-22

Report

This manuscript is a “Community Report,” consisting of a collection of contributions based on the activities developed during the Nordita Program “Quantum Gravity: from gravitational effective field theories to ultraviolet complete approaches.” The program was structured over four weeks. During the first week, PhD-level lectures were delivered, covering various topics in quantum gravity. Lecture notes summarizing the content of that week are available on arXiv:2412.08690. The remaining three weeks were devoted to workshops focused on different aspects within the broad domain of quantum gravity, ranging from foundational and theoretical issues to phenomenology. In addition to research talks, a significant number of panel discussions were organized during those weeks.

A key feature of the Program was the gathering of different sub-communities within the broader “Quantum Gravity” field. In my view, having an extended period for dialogue among diverse lines of thought is essential in quantum gravity. Challenges faced by one particular approach may be more easily addressed by others, offering insights for progress or revealing common ground between seemingly antagonistic perspectives. Moreover, regardless of the chosen approach, the physical processes to be described are the same. The fundamental question of whether universal properties of quantum gravity emerge across different frameworks can only be answered through intense exchanges of ideas and mutual understanding among sub-communities. On a technical level, different approaches can also benefit from computational tools developed by others. Finally, identifying the precise points where distinct views truly disagree is crucial for progress, as such disagreements can lead to concrete, divergent predictions arising from these theories and thus opening up the chance to rule out some candidates.

The present manuscript compiles summaries of each panel held during the Program, highlighting the key ideas raised by the panelists and participants, along with concluding remarks for each session. This results in a representative overview of both convergences and disagreements among panelists, who largely come from different perspectives within the quantum gravity community. Additionally, all panels were recorded, and links to the recordings are provided alongside each summary, allowing interested readers to follow the full discussions (the lectures from the first week are also available online). The panels featured leading international scientists and experts in quantum gravity and neighboring fields such as quantum cosmology. As such, the material presented here is of exceptional value to the quantum gravity community, particularly for early-career researchers and future generations.
Besides the panel summaries, the manuscript also includes personal statements and reflections from each speaker and panelist, providing their main takeaways from the discussions, as well as their views on promising research directions and urgent problems that deserve attention. In this sense, given that the manuscript is a compilation of discussion summaries and individual perspectives, there is limited scope for questions or suggestions regarding the content itself. I would like to emphasize that the manuscript is very well written and its content is highly innovative, particularly due to the unique format of the Program it documents. It is not a typical proceedings volume summarizing individual talks, but rather a comprehensive collection of discussion summaries and personal views from the speakers and panelists.

I have a few minor suggestions for the authors to consider:
Page 5 (first paragraph): It is stated that “At the quantum level, Einstein’s theory lacks predictivity in the ultraviolet (UV) regime, i.e., at high energies, as it is plagued by UV divergences that cannot be absorbed in a finite number of free parameters.” — In a generic context, I would not object, but since there are several approaches trying precisely to make sense of Einstein’s theory non-perturbatively, the issue of predictivity (or lack of it) is unclear. Hence, I would suggest that the authors emphasize that such a statement is valid within perturbation theory.

Page 5 (end): The phrase “...direct detection of gravitational waves (GWs) [19] and the construction of BH shadows [20]...” — The word “construction” sounds strange to me in the context it was used.

Page 14 (“Landscapes and predictivity” paragraph): The sentence “One key example is QCD: the fundamental Lagrangian is simple, but the low-energy vacuum structure is very complicated.” — I did not understand how this compares (or what is the analogy) with the discussion regarding ST’s landscape. The non-triviality of the vacuum structure in this case is tied to the non-perturbative nature of QCD in the low-energy regime and to my knowledge, the origin of the “landscape problem” in ST is totally different. I would suggest some elaboration on what was meant by this analogy.

Page 19: There is a typo in the third line: “...of new physics, such us…”

I also noticed a few additional minor typos scattered throughout the text, but they are easily fixed and do not need to be listed individually.

In summary: I strongly recommend the publication of this manuscript. I believe the Nordita Program and the resulting material constitute outstanding contributions to the quantum gravity community and to the ongoing effort of building bridges across different approaches within the field.

Recommendation

Publish (meets expectations and criteria for this Journal)

---

## Round 2 · Referee Report · Anonymous (Referee 2) · 2025-6-23

The paper is a summary of the discussions held at the Nordita Program "?Quantum Gravity: from gravitational effective field theories to ultraviolet complete approaches", where different quantum gravity communities engaged in dialog to one another. The basic structure of the document is clear and provides an invaluable resource for Quantum Gravity researchers that often are focused in their own approach and are ignorant about others. I fully agree with referee 1 and undoubtedly recommend publication.

After reading all (amazingly detailed) panel summaries, which clarify all the positions discussed clearly, there is one idea that I got the impression was not very much represented in the meeting: holography. This has been a fundamental corner for many in our field and has led to so much progress on our understanding of Quantum Gravity in the last two decades, so it is unfortunate is not often mentioned in the discussions (only one panelist in one panel makes of it a central role, at least judging from the transcript), and only plays a somewhat central role in a couple of essays. Perhaps this is due to the specific choice of panelists/participants? It would be great if future editions (which I hope are coming) of this wonderful interdisciplinary program can take into account a stronger holographic component.

Next are some comments regarding the way the panel discussions are phrased/reported, or questions for clarification.

- The discussion in panel 2.3 did not clarify what it is meant by the spacetime metric being "fundamental" versus not. To the point in page 12 about gravtions being strongly interacting in the IR, it is perhaps interesting and worth mentioning for context that this is not the case in higher dimensions, where IR divergences at low derivatives are absent.

- In Section 2.4, page 13, I wonder if a rewriting "for overconfident claims made decades ago by ST pioneers" could be rephrased in a less aggressive way without compromising the accuracy of the description of the discussion, e.g. "for any grandiose claims that some ST pioneers might have made decades ago". I think this rewording conveys the same message and I wonder if the original one would really be supported by e.g. a majority of participants of the workshop. Anyway, I for one think that someone being optimistic about their research is a positive trait! and therefore the text could use a more neutral language. That being said, if the panelists really agreed on the literal wording in the text, then I don't see reason to change it.

- Also in this section, page 14, regarding black hole entropy it is said that it is ' a

task where ST claims success'. Why not say "succeeds"? No discussion on the topic is mentioned. I suggest to change it like this or detail the kind of doubts voiced by panelists on this, if any.

- Page 15 "while it is not proven that ST exists non-perturbatively, some evidence exists in some superselection sectors". I think that characterizing all of stringy holography and quantum gravity in AdS (I assume this is what is meant by reference [55]) as "some evidence in some superselection sectors" is not really accurate (see my point above). It is a field of QG research in itself, and the evidence for having a non-perturbative definition for string theory via CFT is overwhelming. That's how I'd describe it; not saying that the authors have to describe it like that, but surely there's a middle ground.

- Page 15: Calling Swampland a "stringy fever dream" might not be the most respectful phrasing for the young colleagues working in this field. Perhaps re-wording to "Swampland: Stringy feature or universal framework" or something like that?

- Section 2.7 might have benefited from contributions of the growing community working on "cosmological correlators", particularly on the S-matrix and cosmology topic of discussion, since there has been much recent progress in the questions addressed there.

- Page 21: "By contrast, in ASQG and other theories, the scale is believed to be the Planck scale. ". Perhaps a short sentence or two could be added explaining why is this expected? Couldnt there be other states below planck in ASQG?

- Page 22: Would it be possible to briefly comment on whether there was any reaction to Brandenberger's statement that inflation may "rest in peace"? This would be hugely controversial statement in large swaths of the community, so it is interesting to learn whether anybody among the panel/participants disagreed or everybody aquiesced with the statement.

- Page 23: Did the panel introduce first why, if there is a breakdown of Lorentz symmetry, it should be related to gravity? Couldnt it be related to some other sector, orthogonal to the QG question? Even if it is common knowledge, a line or two at the beginning regrding this to provide context would be very helpful.

On top of the above, I got two typos:

- Page 5: "Different schools of thought": thought should be singular

- Page 11: "One clear message which came out from the discussion is that there is no agreement on whether spacetime is fundamental, and on the fact that there is no unique way to make spacetime emergent." This sentence is weird; seems to say that there is no agreement "on the fact". But a fact is something not subject to disagreement. Maybe replace fact by question?

---

## Editorial Decision

published